# The contribution of inflammatory astrocytes to BBB impairments in a brain-chip model of Parkinson's disease

A. de Rus Jacquet [1,2,3,10] ✉, M. Alpaugh[1,2,4,10], H. L. Denis [1,2], J. L. Tancredi[3,5], M. Boutin[1], J. Decaestecker[6], C. Beauparlant[6], L. Herrmann[6], M. Saint-Pierre[1], M. Parent[2,7], A. Droit [6], S. Breton[8,9] & F. Cicchetti [1,2] ✉

Astrocyte dysfunction has previously been linked to multiple neurodegenerative disorders including Parkinson's disease (PD). Among their many roles, astrocytes are mediators of the brain immune response, and astrocyte reactivity is a pathological feature of PD. They are also involved in the formation and maintenance of the blood-brain barrier (BBB), but barrier integrity is compromised in people with PD. This study focuses on an unexplored area of PD pathogenesis by characterizing the interplay between astrocytes, inflammation and BBB integrity, and by combining patient-derived induced pluripotent stem cells with microfluidic technologies to generate a 3D human BBB chip. Here we report that astrocytes derived from female donors harboring the PD-related LRRK2 G2019S mutation are pro-inflammatory and fail to support the formation of a functional capillary in vitro. We show that inhibition of MEK1/2 signaling attenuates the inflammatory profile of mutant astrocytes and rescues BBB formation, providing insights into mechanisms regulating barrier integrity in PD. Lastly, we confirm that vascular changes are also observed in the human postmortem substantia nigra of both males and females with PD.

Neurodegenerative disorders such as Parkinson's disease (PD) are characterized by the alteration of neuronal functions and networks, ultimately resulting in neuronal death. A major hallmark of this complex disorder is the loss of dopaminergic neurons in the substantia nigra, and understanding the underlying causes of this selective vulnerability has been a major focus of research. In recent years, pathological features such as abnormal immune activation and blood-brain barrier (BBB) disruption have emerged as potential contributors to neurodegenerative disease onset and/or progression. Abnormal immune activation has been observed in the blood and brains of individuals diagnosed with PD and animal models of the disease[1–4], with evidence of exacerbation of these pathological processes throughout disease progression[5,6]. Previous studies found elevated levels of circulating cytokines[1], as well as features characteristic of reactive astrocytes and microglia that occur early in disease and may precipitate neurodegeneration[7,8]. Astrocyte dysfunction in PD is of

[1]Centre de Recherche du CHU de Québec - Université Laval, Axe Neurosciences, Québec, QC G1V 4G2, Canada. [2]Département de Psychiatrie & Neurosciences, Université Laval, Québec, QC G1V 0A6, Canada. [3]Janelia Research Campus, Howard Hughes Medical Institute, Ashburn, VA 20147, USA. [4]Department of Molecular and Cellular Biology, University of Guelph, Guelph, ON N1G 2W1, Canada. [5]Cell Biology R&D, Thermo Fisher Scientific, Frederick, MD 21704, USA. [6]Centre de Recherche du CHU de Québec - Université Laval, Axe Endocrinologie et Néphrologie, Québec, QC G1V 4G2, Canada. [7]CERVO Brain Research Center, Québec, QC G1E 1T2, Canada. [8]Centre de Recherche du CHU de Québec - Université Laval, Axe Reproduction, santé de la mère et de l'enfant, Québec, QC G1V 4G2, Canada. [9]Centre de recherche en reproduction, développement et santé intergénérationnelle, Université Laval, Québec, QC G1V 4G2, Canada. [10]These authors contributed equally: A. de Rus Jacquet, M. Alpaugh. ✉e-mail: aurelie.jacquet@crchudequebec.ulaval.ca; francesca.cicchetti@crchudequebec.ulaval.ca

particular interest because these unique cells of the CNS are versatile and considered to be master regulators of brain function and homeostasis. They engage in diverse roles, from metabolic coupling with neighboring cells[9] to functions typically associated with the innate immune system, including antigen presentation[10] and secretion of inflammatory mediators[11]. Through their interactions with the brain microvasculature, astrocytes can modulate the entry of circulating factors into the CNS[12], further supporting their importance as regulators of brain function and blood-brain interactions. However, accumulating evidence suggests that astrocytes may fail to successfully perform their protective roles over the course of PD. This includes a failure to provide trophic support for neurons[13,14], an exacerbated neuroinflammatory response[15], and these functional changes may additionally result in compromised BBB integrity[16–18]. Evidence of capillary leakage in PD has been demonstrated in brain imaging studies[17,18], abnormal perivascular deposit of serum proteins was shown using histological analyses of postmortem brain tissue of PD patients[19,20], and albumin/IgG levels were increased in the cerebrospinal fluid of patients compared to controls[21]. In parallel, increased angiogenesis was observed, a process that could lead to the formation of immature blood vessel with weaker blood-brain protective properties[22,23]. The cause of BBB permeability in PD has not been elucidated, but evidence suggests that pro-inflammatory mediators, including cytokines, could affect its integrity[24,25]. In addition, the BBB relies on the coordinated action of brain microvascular endothelial cells (BMECs), pericytes and astrocytes to form a tightly regulated barrier[26], and the complexity of this multi-cellular cytoarchitecture suggests that dysfunction in individual components may impact the function of the whole system and drive pathology.

Previous studies showed that astrocytes generated from patient-derived induced pluripotent stem cells (iPSCs) harboring PD-related mutations mediate non-cell autonomous mechanisms of neurodegeneration. For example, it was found that the LRRK2 G2019S mutation triggers pathological changes to the astrocyte secretome[13,27], resulting in the loss and atrophy of co-cultured dopaminergic neurons[13,14]. The LRRK2 protein has also been proposed to modulate inflammatory responses in vitro[28,29], in vivo[15,30] and in people with PD[31,32], and it has been associated with other biological processes, including vesicle trafficking and mitochondrial function[33,34]. The LRRK2 G2019S mutation is of particular interest as it is linked to both familial and sporadic forms of PD[35–38], and can consequently be used to understand underlying aspects of PD etiology. In this study, we leveraged recent technical advances in the in vitro modeling of the BBB to investigate how astrocyte dysfunction contributes to barrier weakening in PD. We observed that astrocytes harboring the LRRK2 G2019S mutation, but not control astrocytes, have a pro-inflammatory profile and alter vessel morphology and function. Mechanisms regulating the inflammatory phenotype involve the MEK1/2 signaling pathway, and pharmacological inhibition of MEK1/2 activity not only attenuates abnormal cytokine secretion but also rescues vessel integrity. Morphological changes to the brain vasculature observed in the BBB chips are also present in the substantia nigra of PD patients, suggesting that vascular changes likely occur over the course of the disease and may implicate astrocytes.

## Results

### The inflammatory and angiogenic profile of iPSC-derived astrocytes is altered by the LRRK2 G2019S mutation

Previous studies have demonstrated that LRRK2 G2019S astrocytes reduce dopaminergic neuron viability, and RNA-sequencing (RNA-seq) analyses have identified changes to the LRRK2 G2019S astrocyte transcriptome[13,14,27]. To increase the power of these individual studies and potentially reveal new dysregulated pathways, we performed a meta-analysis using three publicly available RNA-seq datasets generated from LRRK2 G2019S mutant or control iPSC-derived astrocytes.

These astrocytes were produced using different protocols and sources of human iPSCs. Two of the differentiation protocols included the neuralization and patterning of iPSC monolayers into midbrain neural progenitor cells (NPC) via dual SMAD inhibition[13,27], and the third protocol relied on the production of NPCs from iPSC-derived neurospheres[14] (Fig. 1A). These three studies used iPSC lines generated independently and originating from different donors, however, each astrocyte line expressed similar levels of astrocyte markers and low to undetectable levels of neuronal markers (Supplementary Fig. 1). After confirming that datasets were comparable, we identified differentially regulated genes between LRRK2 G2019S vs. control astrocytes, and found a similar number of altered genes in cells produced by de Rus Jacquet et al. and di Domenico et al. (2271 and 2337 genes respectively) compared to those generated by Booth et al. (347 genes) (Fig. 1B). To identify biological processes consistently affected across multiple iPSC-derived astrocyte lines and differentiation protocols, we selected the 4008 genes differently regulated in these three datasets (Supplementary Data 1, Supplementary Table 1) and performed a gene ontology (GO) enrichment analysis. This approach revealed previously unidentified upregulated gene sets in LRRK2 G2019S vs. control astrocytes including angiogenesis, a biological process which occurs throughout the life of an individual and is responsible for the formation of new networks of blood vessels[39] (Fig. 1C). Careful analysis of these angiogenesis-related genes revealed that some factors are also associated with regulation of the BBB (e.g. *TBX1*[40], *PTGS2*[41], *SERPINF1*[42], *THBS2*[43], and *FGF10*[44]) (Fig. 1D). Furthermore, we confirmed changes to inflammation-related genes, which is consistent with the proposed role of LRRK2 in modulating immune responses[15,28–30]. Among these inflammation-related genes, a number are also associated with changes to brain vascular function (e.g. *IL6*[45,46], *PTGS2*[47], *CXCL1*[48], *BDKRB1|BDKEB2*[49,50], *JAK2*[51], etc.) (Fig. 1E). Analysis of downregulated genes in LRRK2 G2019S vs. control astrocytes revealed an enrichment in GO terms "axon guidance", "cell adhesion" and "brain development" (Fig. 1F–H). In addition, a number of angiogenesis-related genes were downregulated, including genes involved with BBB regulation (e.g. *TSPAN12*[52], *APOLD1*[53], *EGF*[54]) (Fig. 1G). In parallel, we performed a gene set enrichment analysis (GSEA) and confirmed the over-representation of angiogenesis- and inflammation-related pathways in LRRK2 G2019S vs. control iPSC-derived astrocytes (Supplementary Fig. 2).

To the best of our knowledge, angiogenesis-related processes have not previously been reported to be dysregulated in iPSC-derived LRRK2 G2019S astrocytes. We therefore validated our transcriptomic findings by performing a membrane array to measure secreted levels of angiogenesis-related proteins (Fig. 1I). To this end, we produced iPSC-derived astrocytes as previously described[55] (Supplementary Fig. 3) and collected astrocyte conditioned media (ACM) from LRRK2 G2019S and control cultures. We observed that PD astrocytes differentiated from isogenic and non-isogenic iPSC pairs alter their secretion pattern of molecules known to regulate angiogenesis by promoting vessel formation (e.g. angiopoietin-1[56,57], urokinase-type plasminogen activator (uPA)[58,59], amphiregulin[60]), as well as factors that have been shown to impair the permeability of endothelial vessels (e.g. VEGF)[61,62] (Fig. 1J). These findings suggest that LRRK2 G2019S astrocytes may send a combination of pro- and anti-angiogenic signals to endothelial cells which, together with the secretion of inflammatory mediators, could alter the formation and maintenance of the brain vasculature.

### Preparation of a BBB-chip model using iPSC-derived cells

Recent studies using iPSC-derived cells have indicated that reactive astrocytes may be involved in vascular changes[63], and clinical studies in PD patients showed evidence of neurovascular dysfunction[16–20]. To determine if astrocytes could contribute to BBB dysfunction in PD, we differentiated BMEC-like cells following a published protocol[64,65] and established a human iPSC-based 3D model that recapitulates the

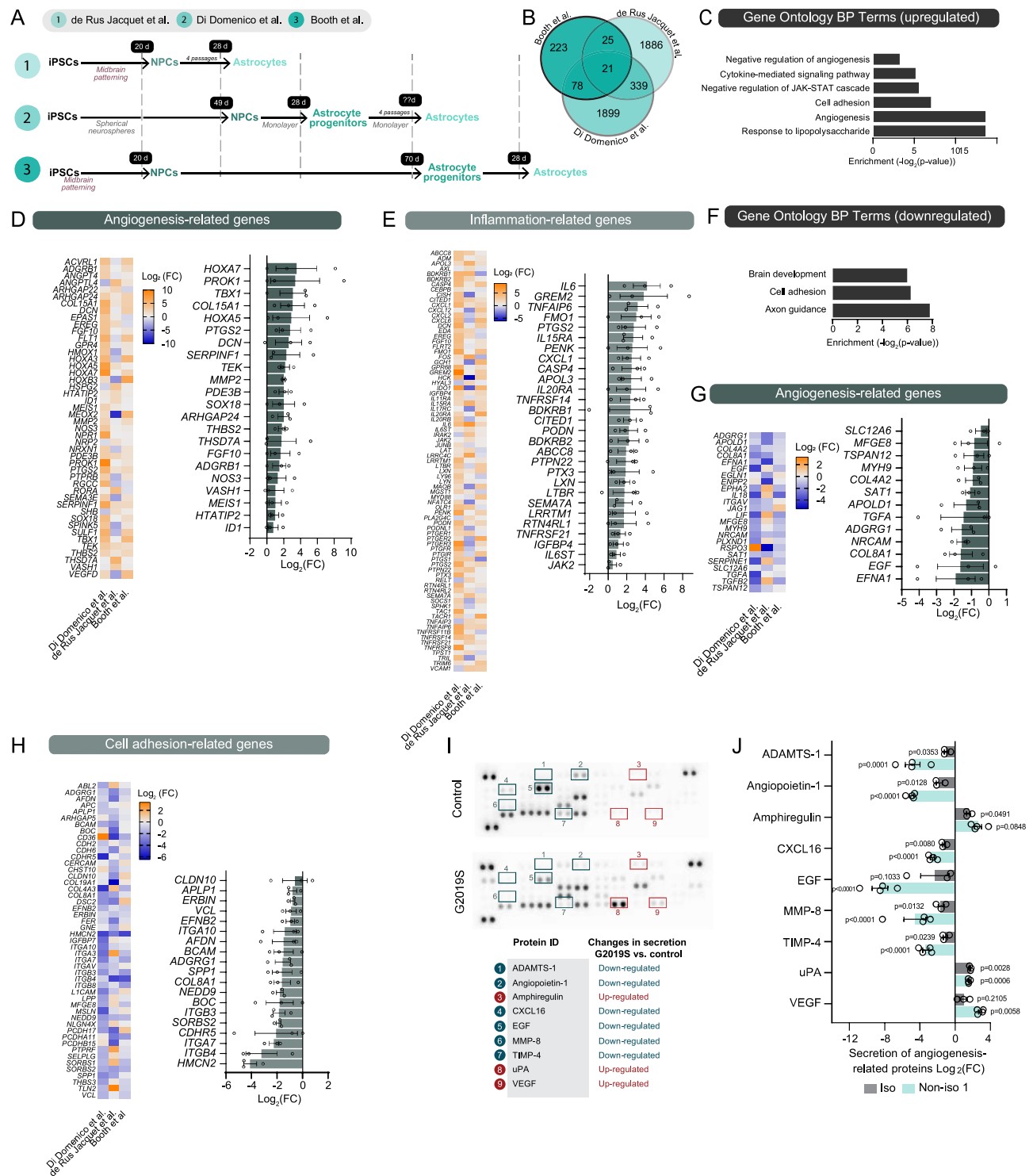

unique cytoarchitecture and microenvironment of the neurovascular unit (NVU) (Fig. 2A). Specifically, a 3-lane microfluidic chip was used to enable the co-culture of an endothelial-like vessel in the top lane (i.e. vascular compartment) and pericytes and astrocytes in the bottom lane (i.e. brain compartment). This model also recreates perfusion conditions that mimic in vivo shear forces which contribute to the migration and re-organization of BMECs into a fully formed endothelial vessel[66]. The vascular compartment is coated with collagen IV and fibronectin to recapitulate features of the basement membrane[67]. In addition, it directly faces a semi-rigid collagen I extracellular matrix (ECM) that provides a scaffold for vessel formation while allowing the passive diffusion of secreted molecules from the top (endothelial) to

the bottom (glial) lane, and vice versa. Analysis of gene expression in BMEC-like cells produced from two different control iPSC lines confirmed the increased expression of benchmark endothelial markers, including *CDH5* (coding for the tight junction protein VE-cadherin) and *ABCB1* (coding for the efflux transporter p-glycoprotein) (Fig. 2B). Further analysis of tight junction proteins by western blot (Fig. 2C) and immunofluorescence (Fig. 2D, E) validated the presence and cellular localization of these proteins in BMEC-like cells. Notably, they express similar levels of VE-cadherin and claudin 5 compared to human primary BMECs, confirming the presence of endothelial features in these cells. We also found that these cells express a number of epithelial makers (Supplementary Fig. 4A), supporting findings from previous

**Fig. 1 | Meta-analysis of RNA-sequencing data reveals transcriptomic changes in inflammation and angiogenesis-related processes. A** Summary of the differentiation protocols utilized in the three studies included in the analysis to derive astrocytes from iPSCs. **B** Venn diagram illustrating the number of overlapping genes between each RNA-sequencing (RNA-seq) dataset. **C** Gene ontology analysis show upregulated components identified by RNA-seq. Benjamini–Hochberg adjusted p-values were obtained from the Database for Annotation, Visualization and Integrated Discovery (DAVID) tool. **D, E** Heatmaps representing the differential expression of genes encoding angiogenesis- (**D**) or inflammation-related factors (**E**) in LRRK2 G2019S vs. control iPSC-derived astrocytes. Log$_2$(FC) represents disease vs. control fold change in gene expression. Histograms show the log$_2$(FC) values of a selection of genes sorted in order of descending fold-change. **F** Gene ontology analysis of downregulated components identified by RNA-seq. **G, H** Heatmaps representing the differential expression of genes encoding angiogenesis- (**G**) or cell adhesion-related factors (**H**) in LRRK2 G2019S vs. control iPSC-derived astrocytes. Histograms report the log$_2$(FC) values of a selection of genes sorted in order of descending fold-change. **I** Representative images of angiogenesis membrane arrays obtained using control or LRRK2 G2019S astrocyte conditioned media. Each dark dot represents a specific angiogenic factor spotted in duplicate onto the membrane. The table shows a selection of proteins either over-secreted (red) or under-secreted (blue) by iPSC-derived LRRK2 G2019S vs. control astrocytes, and the numbers refer to specific spots on the membranes. All secreted proteins were quantified and shown in panel **J**. **J** Histogram reporting the membrane quantification of secreted angiogenic factors by iPSC-derived LRRK2 G2019S vs. control astrocytes for an isogenic and non-isogenic pair. RNA-seq data were obtained from two (Di Domenico et al.), three (de Rus Jacquet et al.) or four (Booth et al.) biological replicates; Data in (**D**, **E**, **G**, **H**) represent the differential gene expression calculated for each independent RNA-seq study (n = 3 biological replicates); angiogenesis array data in (**J**) are from three (isogenic line) and four (non-isogenic line) independent biological replicates; error bars represent mean + standard error of the mean (SEM). Statistical analysis was performed prior to log$_2$ transformation using one sample t test with a theoretical mean of 1 (**J**). iso isogenic iPSC line, non-iso non-isogenic iPSC line. Source data are provided as a Source Data file.

studies[65,68]. Newly generated BMEC-like cells, iPSC-derived astrocytes and primary human pericytes were plated into their respective lanes and allowed to grow and freely migrate into the middle ECM lane (Fig. 2E). After 6 days in vitro (DIV), barrier integrity was assessed by calculating apparent permeability ($P_{app}$) coefficients of the vessel using fluorescein and 4.4 kDa dextran-TMRE dyes, as well as the p-glycoprotein substrate rhodamine. Our data show that this model enables the formation of vessels that prevent the migration of fluorescein and 4.4 kDa dextran-TMRE into the brain compartment (Fig. 2F, Supplementary Fig. 4B), thus demonstrating the successful formation of functional tight junctions. In addition, quantification of rhodamine retention in the vessel suggests effective p-glycoprotein-mediated efflux activity in the newly formed capillaries. Furthermore, the 3D vessels regulate the passage of IgG into the brain compartment, hence recapitulating an important feature of the healthy brain[69] (Fig. 2G). However, BMEC-like cells fail to produce leaktight vessels when the brain compartment is seeded with pericytes and without astrocytes (Fig. 2H, I), but astrocytes alone are sufficient to promote functional vessel formation (Fig. 2J, K, Supplementary Fig. 4C), thus supporting the relevance of our model to study the role of astrocytes in vessel formation and function.

### IPSC-derived astrocytes with the LRRK2 G2019S mutation do not support the formation of a functional vessel

To understand the specific effects of dysfunctional astrocytes on vessel formation, we introduced either control or LRRK2 G2019S astrocytes into our BBB model. We produced BBB chips consisting of control BMEC-like cells, control pericytes and astrocytes (BBB$^{CTL}$) and compared barrier integrity with chips containing control BMEC-like cells, control pericytes and LRRK2 G2019S astrocytes (BBB$^{G2019S}$). We performed these experiments using three independent iPSC pairs prepared from different PD patients with the LRRK2 G2019S mutation. Two patient lines were paired with iPSCs generated from a healthy sex- and age- matched individual (non-isogenic pairs)[13]. A third mutant line was genetically-corrected to produce a control line that differs exclusively at the point mutation (isogenic pair)[70]. The distinct advantage of this isogenic pair is the elimination of confounding factors associated with differences in genetic backgrounds that could influence the experimental results.

When isogenic and non-isogenic pairs were compared, the presence of mutant astrocytes increased leakage of 4.4 kDa dextran-TMRE and rhodamine through the endothelial-like vessel (Fig. 3A). Measurement of $P_{app}$ values show a main effect of astrocyte genotype on the leakage of 4.4 kDa dextran-TMRE (LRRK2 G2019S: $F_{1,24} = 21.74$, $p < 0.0001$) (Fig. 3B), fluorescein (LRRK2 G2019S: $F_{1,12} = 98.63$, $p < 0.0001$) (Supplementary Fig. 4D), and rhodamine (LRRK2 G2019S: $F_{1,21} = 16.24$, $p = 0.0006$) (Fig. 3C) in BBB$^{G2019S}$. For rhodamine, an effect

of iPSC lines was also present suggesting that even within disease groups, some variability was observed (iPSC lines: $F_{2,21} = 4.167$, $p = 0.0299$). Confocal imaging, followed by 3D reconstruction and cross-sectioning of the vessels, revealed areas of low cell density in BBB$^{G2019S}$ but not BBB$^{CTL}$ (Fig. 3D). The presence of regions of low cell density raised the possibility that the changes in $P_{app}$ values could be secondary to disrupted vessel formation. To test this possibility, we performed a transwell assay in which monolayers of control BMEC-like cells were plated in the top (i.e vascular) chamber, and control pericytes with either control or LRRK2 G2019S astrocytes were plated in the bottom (i.e. brain) chamber. Using this simplified model, we confirmed that mutant astrocytes significantly increase endothelial permeability in monolayers of BMEC-like cells (Fig. 3E).

To better understand the mechanisms surrounding barrier disruption, we performed western blots and quantified tight junction protein levels for VE-cadherin, ZO-1 and claudin 5 in the microfluidic model. In contrast to our barrier integrity data, none of these proteins revealed differences between BBB$^{G2019S}$ and BBB$^{CTL}$ (Fig. 3F–I). However, measurement of BMEC-like cell area indicated that vessel-forming cells in BBB$^{G2019S}$ were enlarged as compared to BBB$^{CTL}$ (Fig. 3J–L) and that there was an overall increased vessel width in the mutant BBB chips (Fig. 3M). Overall, these findings suggest that LRRK2 G2019S astrocytes alter the morphology and function of BMEC-like cells while also failing to support their 3D reorganization into a functional vessel, in a manner that is independent of tight junction protein levels.

### Pro-angiogenic signals partially rescue LRRK2 G2019S mediated BBB impairments

To identify potential astrocyte-secreted factors involved in the impairment of BBB function, we turned to our transcriptomic and protein array analyses and focused on altered angiogenesis-related factors. Notably, our data report the dysregulation of *APOL3* (Fig. 1E) and *APOLD1* (Fig. 1G), two genes coding for members of the apolipoprotein family, suggesting that astrocytes with the LRRK2 G2019S mutation present with changes related to this broad family of signaling molecules. Within the large group of apolipoproteins, apolipoprotein E (ApoE) is one of the most studied members due to its implications in Alzheimer's disease. Previous studies found that loss of ApoE3 results in BBB leakage[71], we therefore evaluated the secretion of this protein by LRRK2 G2019S astrocytes. We observed a significant reduction in ApoE release in the conditioned media prepared from LRRK2 G2019S vs. isogenic control astrocytes, in both monolayers and BBB chips (Supplementary Fig. 5A, B). We therefore assessed whether supplementing the brain compartment with ApoE could restore barrier integrity, and found a significant reduction of passive paracellular permeability at day 4 which was lost by day 6 (Supplementary Fig. 5C, F), suggesting that other mechanisms are driving the phenotype.

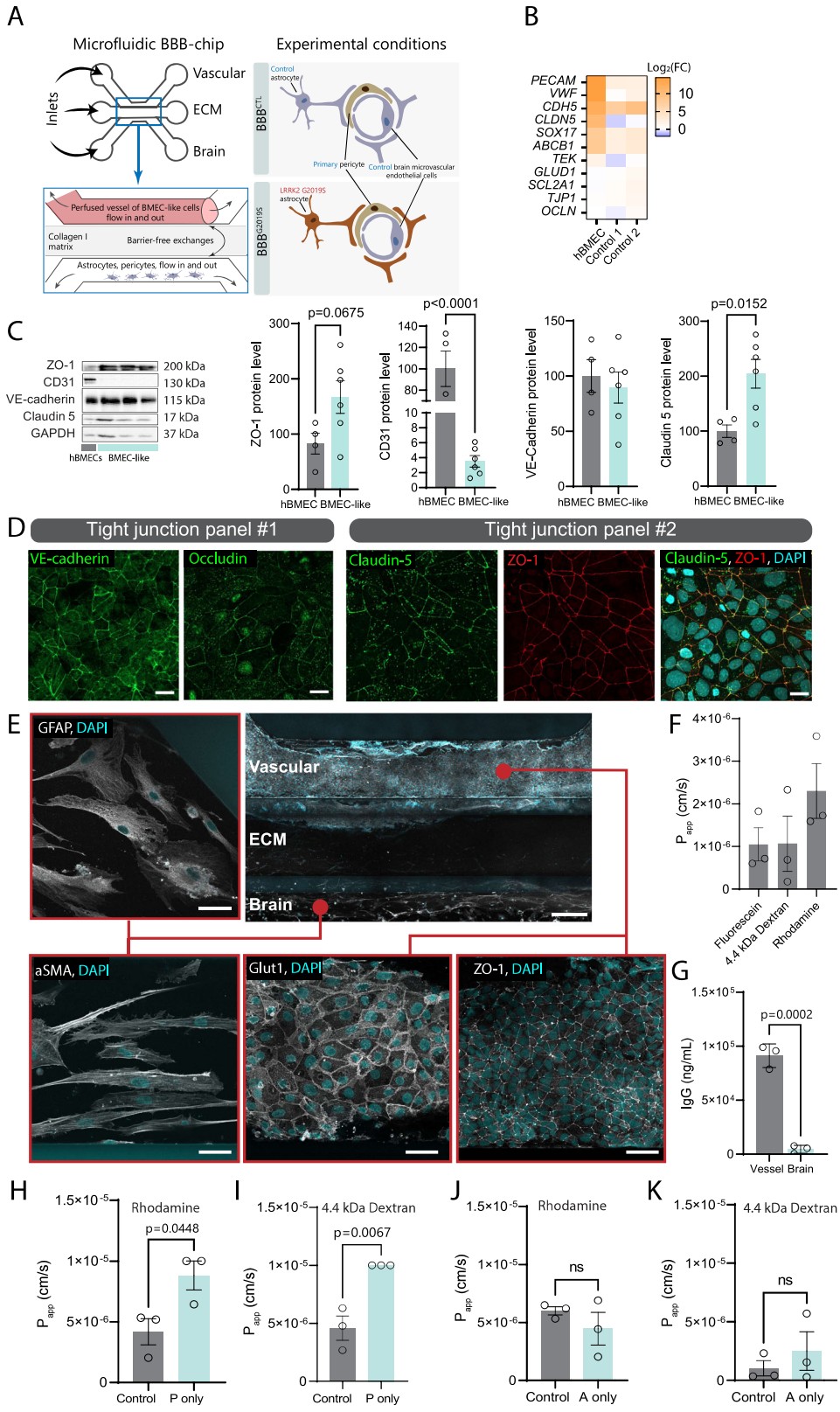

To further understand the contribution of diminished angiogenic factors, we next attempted to rescue BBB$^{G2019S}$ vessel function by activating potent Wnt-related angiogenic pathways in the forming vessels. To this end, BBB$^{G2019S}$ chips were incubated with CHIR99021, a small molecule activator of Wnt signaling, and barrier permeability to 4.4 kDa dextran-TMRE and rhodamine tracers was quantified. Data revealed that Wnt activation in the vessels rescued long-term paracellular (Supplementary Fig. 5G, H), but not transcellular (Supplementary Fig. 5I, J) permeability. These findings imply that increasing pro-BBB signaling pathways is not sufficient to sustain long-term vessel function in BBB$^{G2019S}$ chips, suggesting that mutant astrocytes may secrete detrimental factors that lower BBB integrity.

**Fig. 2 | Establishing a human model of the BBB using iPSC and microfluidic technologies. A** Model showing the BBB chip and experimental design. Inlet arrows indicate where the cells and ECM gel are loaded. **B** Gene expression validation by RT-qPCR of markers of endothelial cell identity in primary human brain microvascular endothelial cells (hBMEC) or BMEC-like cell monolayers. The heatmap represents the $\log_2$(fold change) values of hBMEC or control BMEC-like cells vs. control iPSCs. **C** Western blot-based quantification of tight junction proteins ZO-1, CD31, VE-cadherin and claudin 5 in hBMEC and BMEC-like cell monolayers normalized to GAPDH loading control. Data are represented as the combined protein levels for three different control iPSC lines independently differentiated into BMEC-like cells. **D** Confocal images of immunostained control BMEC-like cells illustrate the expression and localization of tight junction proteins VE-cadherin (green), occludin (green), claudin-5 (green), ZO-1 (red) and merged claudin-5 (red) and ZO-1 (green) with the nuclear marker DAPI represented in blue. **E** Confocal images of an immunostained BBB chip indicate expression of ZO-1 and Glut1 (white, bottom center and right panels), and GFAP and αSMA (white, left panels). Blue structures in all images represent the nuclear marker DAPI. **F** Retention of fluorescein, 4.4 kDa dextran-TMRE and p-glycoprotein substrate rhodamine over a 40-min incubation in BBB$^{CTL}$ vessels cultured for 6 days in vitro (6 DIV). **G** Graphs showing BBB$^{CTL}$ vessel permeability to IgG. **H–K** Graphs showing vessel permeability to rhodamine (**H, J**) and 4.4 kDa dextran-TMRE (**I, K**) at 6 DIV when a BBB$^{CTL}$ is prepared with pericytes only (P only) in the absence of iPSC-derived astrocytes (**H, I**), or with astrocytes only (A only) (**J, K**). Data are from three (**B, F–K**) biological replicates; in (**C**), data were produced using a total of six biological BMEC-like cell replicates originating from three independent iPSC lines. Error bars represent mean + SEM. Statistical analysis was performed using two-tailed unpaired Student's $t$ test with equal standard deviation (s.d.), Scale bars: 20 μm (**D**), 50 and 200 μm (**E**). The BBB$^{CTL}$ and BBB$^{G2019S}$ nomenclature refers to the presence of either control or LRRK2 G2019S astrocytes in the brain compartment of the BBB chip. $P_{app}$ apparent permeability, s seconds. Source data are provided as Source Data file.

## The LRRK2 G2019S mutation alters the inflammatory profile of iPSC-derived astrocytes and reduces barrier integrity

In our transcriptomics dataset, inflammatory pathways were as strongly enriched as those related to angiogenesis. Furthermore, clinical studies have previously reported abnormal levels of cytokines in the blood and brains of patients with PD[1,72,73], while inflammation has been suggested to affect BBB function[24,63,74,75]. We therefore investigated whether the LRRK2 G2019S mutation alters the production and release of inflammatory mediators by iPSC-derived astrocytes. Previous studies defined three reactive astrocyte states, namely A1, A2 and pan-reactive, based on their expression of specific markers[7,76]. We first confirmed that our differentiation protocol results in non-reactive astrocytes that only express negligible levels of these gene markers, despite the presence of fetal bovine serum (FBS) in the growth medium, and those gene expression levels are comparable to astrocytes produced in FBS-free medium (e.g. study by Booth et al.[27]) (Supplementary Fig. 6). We then measured the expression level of these markers in iPSC-derived astrocytes by RT-qPCR and found that the LRRK2 G2019S mutation is associated with an upregulation of A1, A2 and pan-reactive genes in isogenic and non-isogenic astrocytes (Fig. 4A). Additional quantification of an inflammasome component (*NLRP3*), adhesion molecules (*ICAM1, VCAM1*) and inflammatory cytokines (*IL6, CXCL8*) showed a significant upregulation in mutant vs. control iPSC-derived astrocytes (Fig. 4B). ELISA-based quantification revealed concentrations reaching ~150 (isogenic pair) to 500 (non-isogenic pair) pg/mL for IL-6 (Fig. 4C) and ~1400 (isogenic pair) to 4000 (non-isogenic pair) pg/mL for IL-8 (Fig. 4D). These results suggest that the LRRK2 G2019S mutation affects astrocyte signaling by maintaining the cells in a reactive state via up-regulation of gene expression which ultimately alters the secretome.

Inflammatory molecules such as IL-6 and IL-8 have been suggested to alter neurovascular functions[24,65,75,77] and could explain, at least in part, the increased vessel permeability observed in BBB$^{G2019S}$. To determine if these cytokines alone can induce a loss of barrier integrity similar to that of BBB$^{G2019S}$, we treated BBB$^{CTL}$ with IL-6 and IL-8 and measured barrier integrity after 6 days in culture. We observed a significantly increased permeability to the p-glycoprotein substrate rhodamine (Fig. 4E), and neutralization of IL-8 partially rescued BBB$^{G2019S}$ paracellular permeability (Fig. 4F). This suggests that inflammatory mediators known to be secreted by mutant astrocytes could contribute to BBB$^{G2019S}$ dysfunction, and the partial rescue of barrier integrity by an IL-8 neutralizing antibody implies that other molecules may act synergistically.

## Pharmacological inhibition of MEK1/2 phosphorylation rescues the pro-inflammatory phenotype of LRRK2 G2019S astrocytes

Neither IL-8 neutralization or increased pro-angiogenic signals were able to completely reverse the detrimental effects of LRRK2 G2019S astrocytes. Therefore, in our search for molecular targets capable of mediating both the inflammatory- and angiogenic-related responses, we performed a computational analysis which suggested that transcription factors downstream of the ERK cascade target differentially expressed genes (DEG) belonging to the GO categories of angiogenesis, inflammation and cell adhesion. Notably, the top 2 ERK-related transcription factors identified in this analysis (i.e. *MECOM* and *KLF6*) target 55 to 85% of the DEG across the three GO categories. Therefore, this data positions the MEK/ERK pathway as a strong candidate implicated in BBB$^{G2019S}$ dysfunction (Fig. 5A). The role of the MEK/ERK cascade (Fig. 5B) in regulating the production of inflammatory cytokines has been previously described[78–80], and it has been reported that ERK1/2 mediates pathological changes in LRRK2 G2019S dopaminergic neurons[70]. Together with our computational analysis, these studies suggest that the MEK/ERK signaling pathway may also be dysregulated in iPSC-derived LRRK2 G2019S astrocytes. Western blot quantification of phosphorylated ERK1/2 (p-ERK1/2), the active form of the protein, showed an increase in mutant astrocytes compared to control (Fig. 5C), and inhibition of the pathway using the selective MEK1/2 inhibitor PD0325901 reversed expression of A1, A2 and pan-reactive genes in isogenic and non-isogenic astrocytes (Fig. 5D). The small molecule also reduced expression of *NLRP3*, *IL6* and *CXCL8* (Fig. 5E). We observed a similar decrease in *NLRP3* and *CXCL8* gene expression when mutant astrocytes were treated with the ERK1/2 inhibitor SCH772984, but *IL6* levels remained unchanged (Fig. 5F). This observation suggests that other signaling pathways may regulate *IL6* overexpression in LRRK2 G2019S astrocytes. We then assessed whether MEK1/2 inhibition would result in reduced secretion of inflammatory cytokines by PD astrocytes and observed that PD0325901 treatment significantly down-regulated IL-8 release in ACM prepared from isogenic or non-isogenic (Fig. 5G) cultures. However, the treatment had only modest effects on IL-6 release, which was slightly decreased in non-isogenic, but not in isogenic cultures (Fig. 5H). This finding further supports the possibility that alternative signaling pathways may regulate IL-6 levels.

## Impaired formation of functional endothelial-like vessels by LRRK2 G2019S astrocytes is mediated by MEK1/2 phosphorylation

Our data suggesting that inhibition of MEK/ERK signaling significantly rescued the inflammatory phenotypes of LRRK2 G2019S astrocytes, we next wanted to determine if this attenuation of the inflammatory response could also improve barrier integrity. We treated the brain compartment of a BBB$^{G2019S}$ with the MEK1/2 inhibitor PD0325901 and confirmed decreased IL-8 and IL-6 release into the culture media collected from non-isogenic and isogenic (Fig. 6A, B) BBB chips. Confocal imaging of inhibitor-treated BBB$^{G2019S}$ revealed fully formed vessels (Fig. 6C) that efficiently retained rhodamine (Fig. 6D, E) and 4.4 kDa dextran-TMRE (Fig. 6F) in both isogenic and non-isogenic BBB chips. We also observed that the small

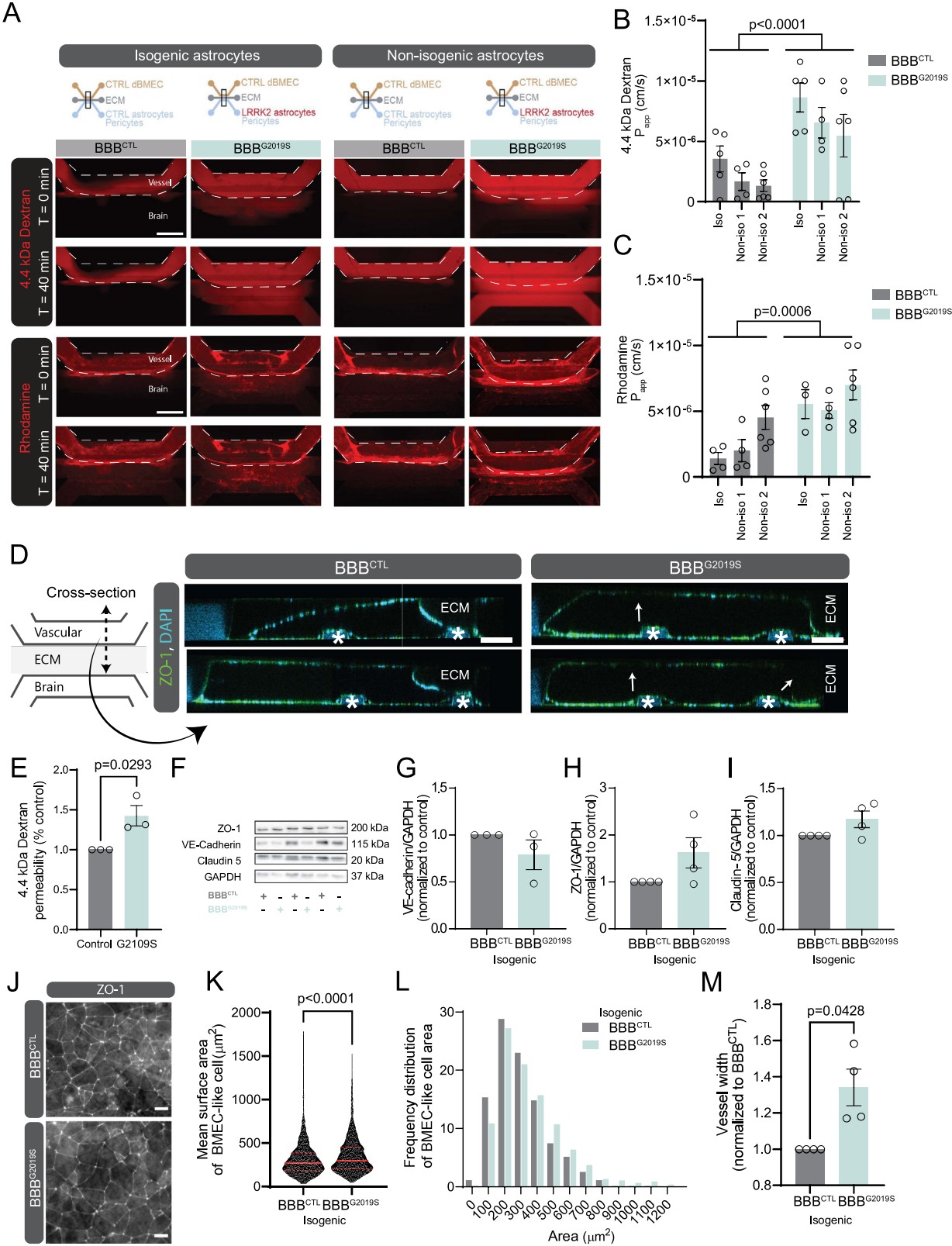

molecule treatment preserved vascular morphology by reducing vessel width (Fig. 6G). In parallel, we aimed to determine whether LRRK2 kinase activity also mediates BBB$^{G2019S}$ loss of vessel integrity. To this end, the brain compartment was treated with 5 μm LRRK2-IN-1, a small molecule kinase inhibitor specific to the LRRK2 protein. We found that LRRK2-IN-1 treatment was not sufficient to ameliorate barrier integrity (Fig. 6H, I) and this observation could be explained

by the limitations of a pharmacological approach. Treatment of the brain compartment did not specifically target the mutant astrocytes, and LRRK2 inhibition also affected pericytes and the abluminal part of the vessel wall. Given the importance of LRRK2 in cell-cell communication[13], a broad, non-targeted LRRK2 inhibition appears inefficient to restore barrier formation. Furthermore, LRRK2 kinase activity is likely to impact the early stages of astrocyte differentiation,

**Fig. 3 | Astrocytes with the LRRK2 G2019S mutation fail to support the formation of a functional BBB. A** Representative images of 4.4 kDa dextran-TMRE (red, top images) or rhodamine (red, bottom images) in BBB$^{G2019S}$ vs. BBB$^{CTL}$ chips after a 40-min incubation, prepared using two independent LRRK2 G2019S iPSC pairs. **B, C** Quantification of 4.4 kDa dextran-TMRE (**B**) and rhodamine (**C**) apparent permeability ($P_{app}$) values in chips prepared using three independent iPSC pairs. **D** Confocal images illustrating two cross-sections of immunostained vessels expressing ZO-1 (green) and DAPI nuclear stain (blue). White arrows indicate areas of low or absent immunoreactivity, white stars point to the localization of guides imprinted on the microchip. **E** Quantification of 4.4 kDa dextran-TMRE permeability in a transwell system, results for isogenic and non-isogenic line 1 are combined into a single graph. **F** Representative immunoblots depicting protein levels of tight junction markers VE-cadherin, ZO-1, claudin-5 and loading control GAPDH in vessels. **G–I** Quantifications of VE-cadherin (**G**), ZO-1 (**H**), and claudin-5 (**I**) protein levels in vessels were normalized to GADPH. Data are shown as the fold change of BBB$^{G2019S}$ levels compared to BBB$^{CTL}$. **J** Representative images of immunostained BBB$^{CTL}$ and BBB$^{G2019S}$ vessels depicting ZO-1 expression in BMEC-like cells. **K, L** Quantification of control and LRRK2 G2019S BMEC-like cell mean surface area (**K**) and frequency distribution (**L**). **M** Graph representing vessel width in BBB$^{G2019S}$ chips normalized to BBB$^{CTL}$. Data are from three (**E, G**), four (**H, I, M**), five (**B**) or six (**C**) biological replicates; data in (**K, L**) are sampled from >900 individual cells from three independent biological replicates; error bars represent mean + SEM. Outliers were identified using Grubbs' test with an alpha value set at 0.05 and removed from analysis. Statistical analysis was performed using two-tailed unpaired Student's *t* test with equal s.d. Violin plot in (**K**) shows the median (red line) and quartile (red dotted line) values. Scale bar: 600 μm (**A**), 100 μm (**D**), 15 μm (**J**). The BBB$^{CTL}$ and BBB$^{G2019S}$ nomenclature refers to the presence of either control or LRRK2 G2019S astrocytes in the brain compartment of the BBB chip. cm centimeter, iso isogenic iPSC line, kDa kilodalton, min minutes, non-iso non-isogenic iPSC line, s seconds. Source data are provided as a Source Data file.

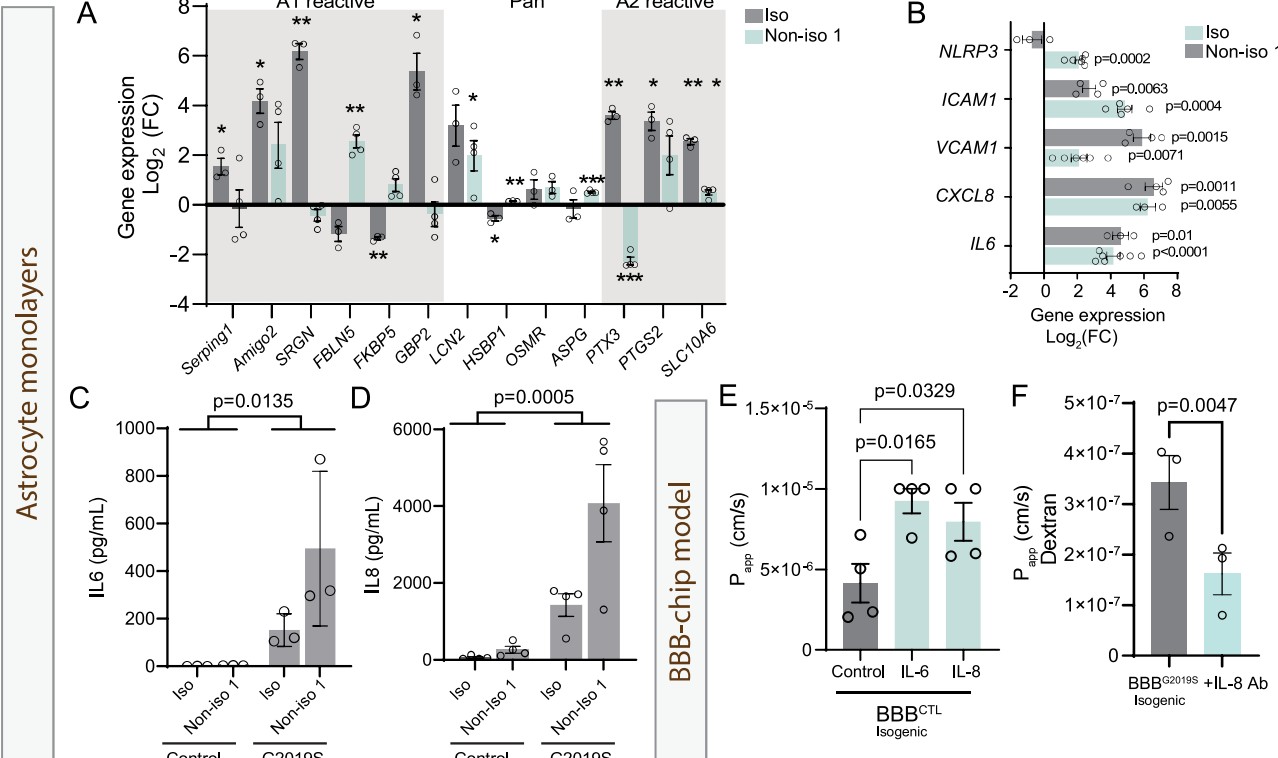

**Fig. 4 | Astrocytes with the LRRK2 G2019S mutation are pro-inflammatory.**
**A** Gene expression validation by RT-qPCR of a panel of 13 markers characteristic of astrocyte reactivity, using control and LRRK2 G2019S astrocytes cultured in monolayers. Data are shown as log$_2$(fold change) values of LRRK2 G2019S vs. control astrocytes for the isogenic and non-isogenic pairs. **B** Gene expression quantification by RT-qPCR of the inflammasome component *NLRP3*, adhesion molecules *ICAM1* and *VCAM1*, and pro-inflammatory cytokines *CXCL8* and *IL6* for the isogenic and non-isogenic astrocyte pairs grown as monolayers. Data are shown as log$_2$(fold change) values of LRRK2 G2019S vs. control astrocytes. **C, D** ELISA-based quantification of IL-6 (**C**) and IL-8 (**D**) concentration in astrocyte conditioned media prepared from isogenic and non-isogenic pairs. **E** Graph showing vessel permeability to rhodamine at 6 days in vitro (DIV) when the brain compartment of a BBB$^{CTL}$ is treated with 100 ng/mL IL-6 or IL-8 for 6 days. **F** Graph showing vessel permeability to 4.4 kDa dextran-TMRE when the brain

compartment of a BBB$^{G2019S}$ chip is treated with 0.8 μg/mL IL-8 neutralizing antibody. Data are from three (**A**, iso; **B**, iso *NLRP3*, non-iso *CXCL10*, iso *IL6*; **C, F**), four (**A**, non-iso; **B**, iso *ICAM1*, iso *VCAM1*, non-iso *CXCL10*; **D, E**), five (**B**, iso *ICAM1*), six (**B**, non-iso *NLRP1* and *VCAM1*), or seven (**B**, non-iso *IL6*) biological replicates; error bars represent mean + SEM. Statistical analysis was performed using one sample *t* test with a theoretical mean of 0 (**A–C**), two-tailed unpaired Student's *t* test with equal s.d. (**D–G, I**), or a one-way ANOVA with Dunnett's multiple comparisons test (I) (**p ≤ 0.05, **p < 0.01, ***p < 0.001). The exact *p*-values for data shown in (**A**) are available in the source data file. Outliers were identified using Grubbs' test with an alpha value set at 0.05 and removed from analysis. The BBB$^{CTL}$ and BBB$^{G2019S}$ nomenclature refers to the presence of either control or LRRK2 G2019S astrocytes in the brain compartment of the BBB chip. iso isogenic iPSC line, non-iso non-isogenic iPSC line. Source data are provided as a Source Data file.

and these early LRRK2 kinase-mediated changes could affect the later BBB impairments observed in the chip, regardless of kinase inhibition at the time of the experiment. Other studies also reported that pharmacological inhibition of LRRK2 kinase activity does not necessarily attenuate the effects of the LRRK2 G2109S mutation in immune cell types relevant to PD[81].

In light of these observations, MEK/ERK signaling appeared to be a central component regulating astrocyte-vascular interactions in the BBB chip. We therefore measured changes to tight junction levels in BBB$^{G2019S}$ chips prepared in the presence or absence of the MEK1/2 inhibitor PD0325901. Western blot analysis revealed a trend, but not a significant increase in tight junction protein levels (Fig. 7A–C), but we

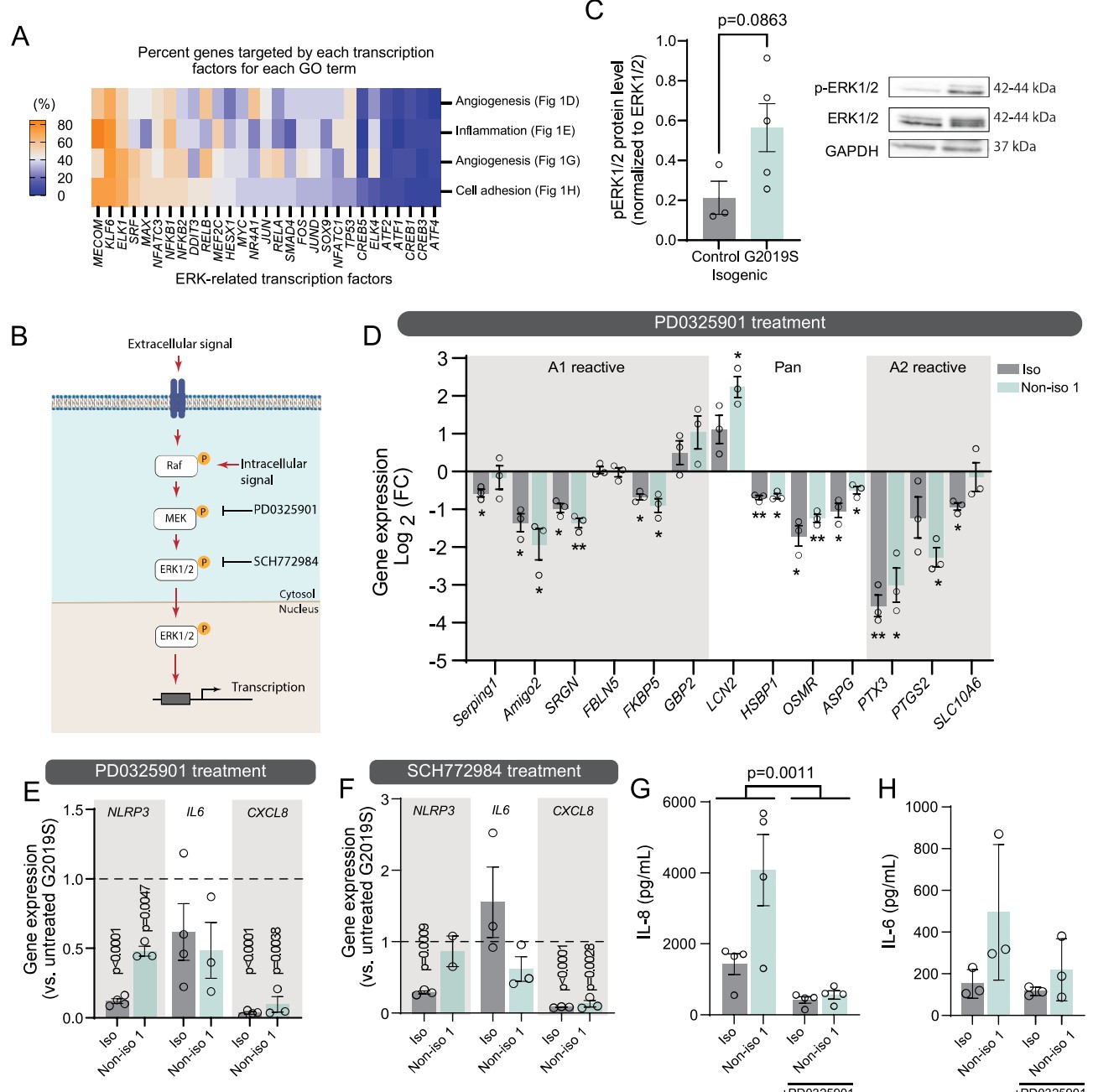

**Fig. 5 | The LRRK2 G2019S mutation is associated with activation of the MEK/ERK pathway, which mediates the astrocyte inflammatory profile. A** Heatmap representing, for each GO term identified in Fig. 1, the percent genes targeted by ERK-related transcription factor. **B** Model showing the MEK/ERK cascade and the targets of small molecule inhibitors PD0325901 and SCH772984 used in this study. **C** Representative immunoblots and quantification of phosphorylated ERK1/2 (p-ERK1/2), total ERK1/2, and loading control GAPDH in LRRK2 G2019S and isogenic control astrocytes cultured as monolayers. **D** Gene expression quantification by RT-qPCR of a panel of 13 markers characteristic of astrocyte reactivity, using control and LRRK2 G2019S astrocytes cultured in monolayers and treated with 0.5 μM PD0325901 for 24 h. Data are shown as $\log_2$(fold change) values of LRRK2 G2019S vs. control astrocytes for the isogenic and non-isogenic pairs. **E, F** Gene expression quantification by RT-qPCR of the inflammasome component *NLRP3* and pro-inflammatory cytokines *CXCL8* and *IL6* for the isogenic and non-isogenic

astrocyte pairs grown as monolayers and treated with 0.5 μM PD0325901 (**E**) or 0.5 μM SCH772984 (**F**) for 24 h. **G, H** ELISA-based quantification of IL-6 (**G**) or IL-8 (**H**) concentration in astrocyte conditioned media prepared from isogenic and non-isogenic pairs treated with 0.5 μM PD0325901 for 24 h. Data are from at least three (**C–F**, **H**) or four (**G**) independent biological replicates; error bars represent mean + SEM. Statistical analysis was performed using one sample *t* test with a theoretical mean of 0 (**D**) or 1 (**E-F**), two-tailed unpaired Student's *t* test with equal s.d. (**C**), or two-way ANOVA with Šídák's multiple comparisons test (**G**) (*$p \le 0.05$, **$p < 0.01$). The exact *p*-values for data shown in (**D**) are available in the source data file. The BBB$^{CTL}$ and BBB$^{G2019S}$ nomenclature refers to the presence of either control or LRRK2 G2019S astrocytes in the brain compartment of the BBB chip. iso isogenic iPSC line, kDa kilodalton, non-iso non-isogenic iPSC line. Source data are provided as a Source Data file.

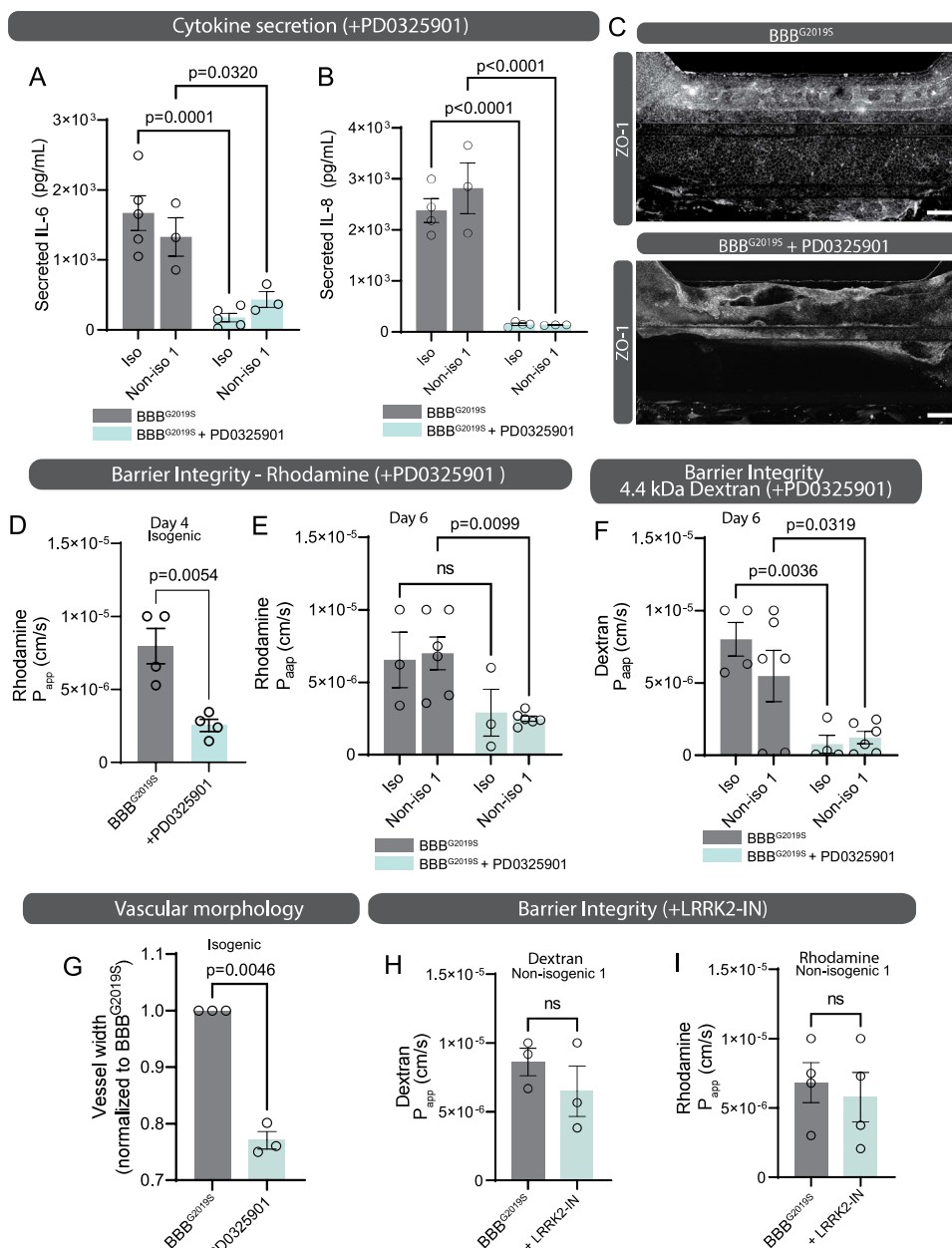

**Fig. 6 | Pharmacological inhibition of MEK1/2 in the brain compartment of the BBB chip rescues barrier integrity. A, B** ELISA-based quantification of IL-6 (**A**) and IL-8 (**B**) in glia conditioned media collected from the brain compartment of isogenic and non-isogenic BBB$^{G2019S}$ chips after treatment with 0.5 μM PD0325901 for 4 to 6 days. **C** Confocal images of BBB$^{G2019S}$ vessels immunostained with tight junction marker ZO-1. These cultures were grown in the absence or presence of 0.5 μM PD0325901 for 6 days. **D–F** Graphs showing vessel permeability to rhodamine (**D, E**) and 4.4 kDa dextran-TMRE (**F**) when the brain compartment of a BBB$^{G2019S}$ chip is treated with regular growth medium or medium supplemented with 0.5 μM PD0325901 for the duration of the experiment. Data were collected using isogenic and non-isogenic iPSC pairs. **G** Quantification of vessel width in BBB$^{G2019S}$ chips

produced in the absence or presence of 0.5 μM PD0325901 for 6 days. **H, I** Graphs showing vessel permeability to 4.4 kDa dextran-TMRE (**H**) and rhodamine (**I**) when the brain compartment of a BBB$^{G2019S}$ chip is treated with regular growth medium or medium supplemented with 5 μM LRRK2-IN-1 for the duration of the experiment. Data are from three (**A, B, E, G, H**), four (**B, D, F, I**), five (**A**) or six (**E, F**) biological replicates; error bars represent mean + SEM. Statistical analysis was performed using two-tailed unpaired Student's *t* test with equal s.d. **A–M**. Scale bar: 200 μm (**C**). The BBB$^{G2019S}$ nomenclature refers to the presence of LRRK2 G2019S astrocytes in the brain compartment of the BBB chip. Source data are provided as a Source Data file.

found greater phosphorylated AKT levels in BBB$^{G2019S}$ treated with PD0325901 compared to untreated vessels (Fig. 7D). This finding suggests that the activation of the pro-health AKT signaling cascade in vessels could mediate at least some of the observed improvements, and is consistent with previous observations of the benefits of AKT activation on BBB integrity[82,83]. In parallel, we evaluated the activation of p38MAPK signaling, a pathway previously associated with the

response of endothelial cells to stress and exogenous stimuli[84]. We found that PD0325901 treatment induced a trend, but not a significant induction of p38 MAPK phosphorylation in BBB$^{G2019S}$ (Fig. 7E). From these results, we conclude that PD0325901 treatment of the brain compartment rescues pathological cytokine secretion in the BBB chip and induces biological processes involved in cellular protection and regulation of vascular processes[85].

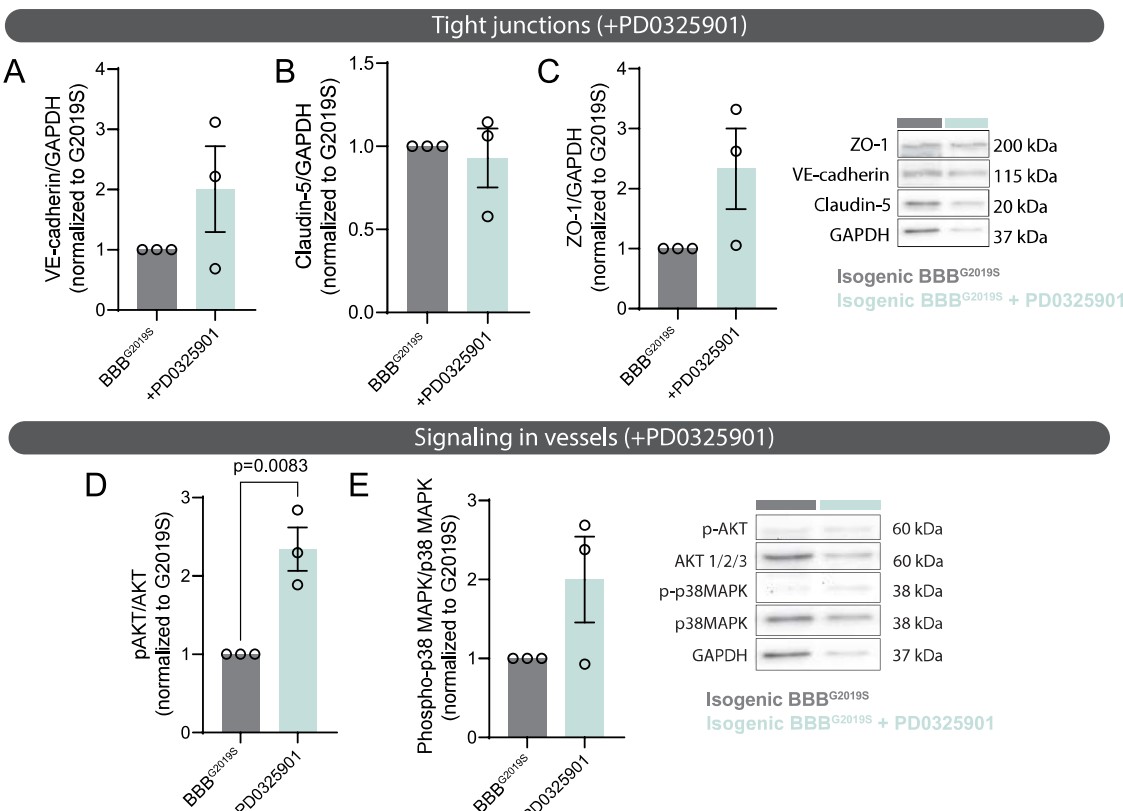

**Fig. 7 | Rescue of BBB function is associated with increased AKT phosphorylation in the vessels. A−C** Immunoblots showing protein levels of tight junction markers VE-cadherin (**A**), claudin-5 (**B**), ZO-1 (**C**), and loading control GAPDH in lysates extracted from the vascular compartment of BBB^G2019S chips in which the brain compartment was treated with regular growth medium or medium supplemented with 0.5 µM PD0325901 for 6 days. Protein levels are normalized to GAPDH loading control and data is shown as the fold change of treated vs. untreated BBB^G2019S vessels. **D**, **E** Immunoblots reporting protein levels of phosphorylated AKT, total AKT (**D**), phosphorylated p38MAPK, total p38 MAPK (**E**), or loading control GAPDH in lysates extracted from the vascular compartment of BBB^G2019S chips in which the brain compartment was treated with regular growth medium or medium supplemented with 0.5 µM PD0325901 for 6 days. Protein levels are normalized to GAPDH loading control and data are shown as the fold change of treated vs. untreated BBB^G2019S vessels. Data are from three biological replicates; error bars represent mean + SEM. Statistical analysis was performed using two-tailed unpaired Student's *t* test with equal s.d. The BBB^G2019S nomenclature refers to the presence of LRRK2 G2019S astrocytes in the brain compartment of the BBB chip. Source data are provided as a Source Data file.

## The PD BBB-chip model recapitulates changes to the vasculature observed in the substantia nigra of PD patients

Establishing a 3D microfluidic model that recapitulates the complex spatial and cellular architecture of the human brain vasculature is an asset to explore neurodegenerative disorders such as PD. The findings from this study suggest that PD-associated mutations significantly affect astrocyte biology and, in turn, vascular morphology and function. Notably, we report the increased diameter of blood vessels in BBB^G2019S chips, a feature that had previously been documented in human postmortem PD brain tissue[86] and that we independently validated in a cohort of five patients (Table 1). Our analysis of postmortem substantia nigra sections confirms an enlargement of laminin-positive vessels from an average 9.25 µm to 10.56 µm (i.e. 12.4% increase) in control vs. PD cases (Fig. 8A−C). As a result, the individual vessel coverage area is increased (Fig. 8D), but further analysis of vessel density revealed a ~25% decrease in the average number of laminin-positive vessels in the substantia nigra (Fig. 8E), accompanied by an overall lower vascular coverage of the diseased tissue (Fig. 8F). Together with previous published work on the loss of BBB function in PD patients, these findings support the concept of vascular abnormalities that progressively develop in the disease.

## Discussion

This study implemented a meta-analysis of previously published RNA-seq datasets, and identified angiogenesis as a biological process potentially modified in a genetic model of PD. This observation was confirmed by measuring changes to the secretion of angiogenesis-related factors in iPSC-derived astrocytes harboring the LRRK2 G2019S mutation. Inclusion of these astrocytes into a 3D model of the BBB revealed impairments in trans and paracellular permeability, further supporting the dysfunction of iPSC-derived LRRK2 G2019S astrocytes. The analysis of human postmortem tissue confirmed that the vascular phenotypes observed in our in vitro system aligned well with changes in the brains of patients with sporadic PD, and these postmortem observations warrant further investigation of astrocyte-vascular interactions in PD. Finally, we were able to show that inhibition of the MEK/ERK pathway was sufficient to rescue barrier deficits raising the possibility that components of this pathway may be a therapeutic target in PD.

BBB formation and maintenance is a tightly regulated process by which cells forming the NVU create a brain microenvironment prone to tissue vascularization[87]. Astrocytes are critically important to the development of this microenvironment and are known to contribute to BBB formation and maintenance via the secretion of signaling molecules and the formation of astrocytic perivascular end feet which surround the capillaries[12,88−90]. BBB dysfunction has been documented in patients with neurodegenerative diseases, including PD[91]. More specifically, clinical studies using medical imaging technologies such as Magnetic Resonance Imaging and Positron Emission Tomography have found evidence of capillary leakage in the brains of patients with PD[17,18]. For example, dynamic contrast enhanced magnetic resonance imaging (DCE-MRI) is a technology developed to measure subtle

changes to BBB integrity, and it was leveraged to identify increased leakage of gadolinium in the basal ganglia of people with PD[17]. A contradicting study, however, reported no difference in the striatal permeability to the potassium analogue rubidium-82 (82Rb), which

was selected to monitor disruption of tight junctions[92]. The authors suggest that their methodological approach may not detect mild changes in 82Rb influx. In another study, investigators utilized positron emission tomography to assess brain infiltration of [11C]-verapamil, a substrate of the efflux transporter p-glycoprotein, and showed increased uptake in the midbrain of people with PD[18]. In addition to imaging studies of barrier integrity, histological analyses of postmortem human brain samples of patients revealed compromised striatal BBB, characterized by abnormal deposits of serum proteins in the brain parenchyma and erythrocyte extravasation[19]. Collectively, these studies support the theory that barrier integrity is reduced in the brains of people with PD. These BBB changes may also be accompanied by angiogenesis-related biochemical changes occurring in the brain of patients, such as increased VEGF[93], which could be involved in vascular remodeling. These collective findings that potentially toxic circulating factors could breach the BBB and enter the brain parenchyma to affect neuronal survival support the hypothesis of a peripheral contribution to PD etiology. To further corroborate this idea, patients undergoing deep brain stimulation of the subthalamic nucleus (STN-DBS) tend to exhibit an attenuation of motor features and slower disease progression, and these important changes correlate with improved brain microvasculature. For example, STN-DBS upregulated ZO-1, claudin 5, VE-cadherin, and occludin levels, as well as decreased microglia density compared to non-STN-DBS patients[20]. On the other side of the barrier, neuroinflammation has been largely described as a pathological event affecting primarily the brain parenchyma, but it is possible

### Table 1 | Demographic details of the cases used in this study

| Age, average (standard deviation) | Control: 68.8 (6.181) PD: 72.40 (5.225) | |
|---|---|---|
| Sex | Male: n = 4 (control), n = 4 (PD) Female: n = 1 (control), n = 1 (PD) | |
| Disease status | Postmortem interval (h) | Cause of death |
| PD | 2.5 | Congestive heart failure and renal failure |
| PD | 17 | Septic shock |
| PD | 14 | Bronchoaspiration, sepsis |
| PD | 7 | Pneumonia |
| PD | 4 | Adenocarcinoma |
| Control | 5 | Lung disorder |
| Control | 20 | Subdural hemorrhage |
| Control | 10 | Myocardial infarction |
| Control | 10 | Myocardial infarction |
| Control | 12 | Aortic rupture |

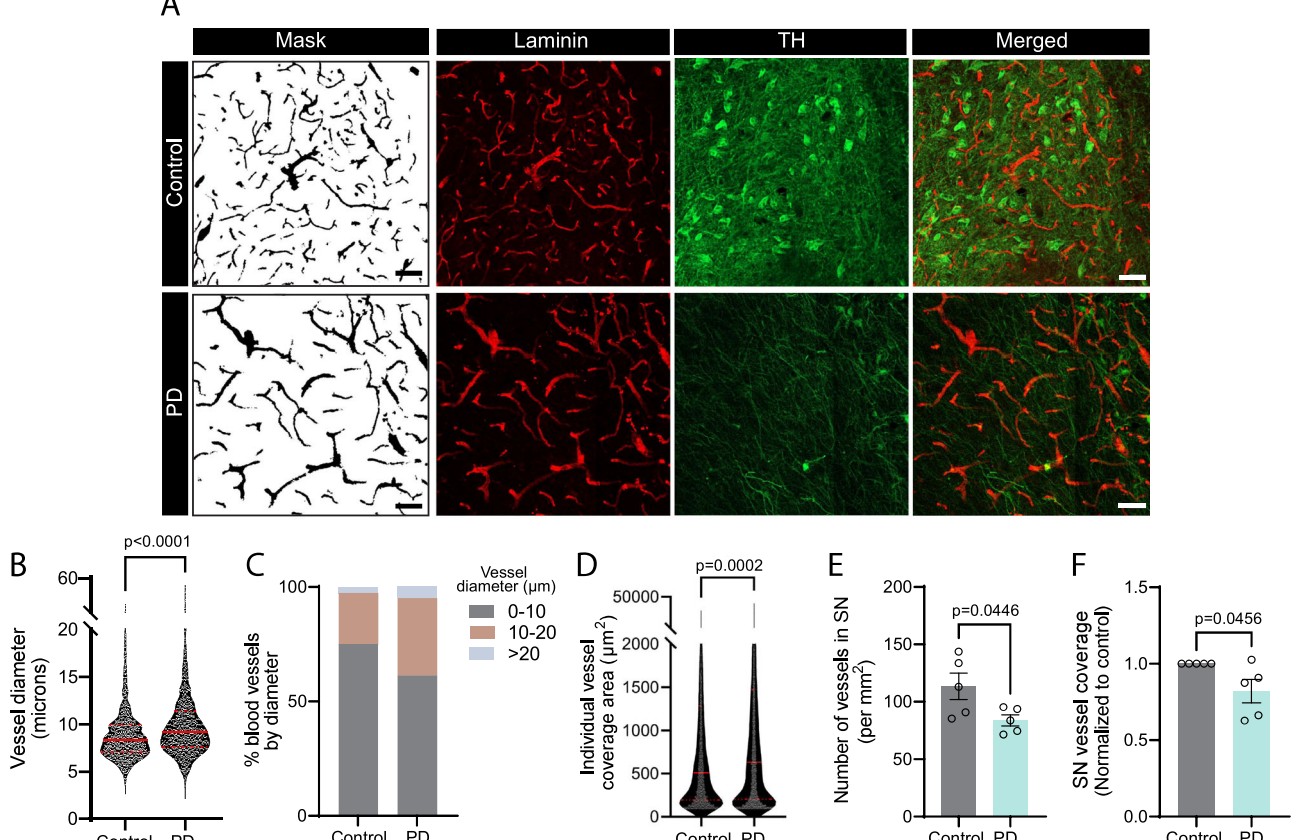

**Fig. 8 | The BBB chip recapitulates morphological changes to the vasculature observed in the substantia nigra of PD patients. A** Confocal images of human postmortem sections of the substantia nigra immunostained for laminin (red) and TH (green). The brain sections were obtained from age- and sex-matched controls or patients with PD. A mask of the laminin-positive staining was produced using FIJI image analysis software. **B–F** Graphs reporting laminin-positive vessel diameter (**B**) and size distribution (**C**), individual laminin-positive vessel coverage area (**D**), number of laminin-positive vessels (**E**), and overall vessel coverage (**F**) in the substantia nigra of PD patients vs control. Data were collected using postmortem brain sections originating from five age- and sex-matched controls and five PD cases; error bars represent mean + SEM. Violin plot in (**B**) and (**D**) indicate the median (red line) and quartile (red dotted line) values. Statistical analysis was performed using two-tailed unpaired Student's t test with equal s.d. Scale bar: 100 μm (**A**). Source data are provided as a Source Data file.

that pro-inflammatory mediators could weaken the BBB and facilitate the entry of blood-borne factors, thus connecting the CNS and periphery into one pathological immune axis. As a consequence, infiltration of peripheral immune cells or plasma-borne cytokines through a breached BBB could exacerbate neuroinflammation, but these processes could also result from an active recruitment by glial cells to assist the brain immune response[94,95]. Astrocytes are immune-competent cells that not only participate in neuroinflammation, but also support BBB function. They may therefore be key regulators of periphery-CNS interactions at the BBB, and could, consequently, affect PD pathology through multiple routes.

Our comparative study of three published RNA-seq datasets, together with biochemical analyses of LRRK2 G2019S astrocytes, revealed changes to angiogenesis- and inflammation-related processes compared to control astrocytes. To determine if PD astrocytes can sustain the formation of an in vitro BBB, we established a model that includes shear flow and allows glia-secreted exogenous cues to reach BMEC-like cells and influence their reorganization into 3D vessels. Evidence of the reliability of this model comes from data demonstrating that (i) formation of leaktight vessels requires the presence of astrocytes (Fig. 2H–K), (ii) barrier integrity is reduced in response to treatment with inflammatory cytokines (Fig. 4E) and (iii) data demonstrating similar changes in size and integrity in the brains of PD patients. The requirement of astrocytes is consistent with previous observations that astrocyte-endothelial interactions are critical to the development and maintenance of a functional brain microvasculature in vivo[12,88].

While our model system offers many advantages, it relies on dyes and the calculation of a $P_{app}$ coefficient in place of the evaluation of transepithelial electrical resistance (TEER). TEER is a widely used method of determining barrier integrity[96], but it relies upon the placement of electrodes on either side of the barrier in order to assess resistance of a cell layer as a readout for tight cell-cell junctions[96]. The BBB-chip platform used here is a micro-scale system made of perfusion channels, and traditional electrodes are not designed for this particular setup. With future advances in technology, assessment of TEER in microfluidic channels may become more accessible. However, the dyes selected for this study remain powerful tools to assess both paracellular (dextran) and transcellular (rhodamine) movements of molecules across the endothelial-like vessel. Using this model system, we were able to compare the barrier properties of BBB^G2019S vs. BBB^CTL, which only differ by the astrocyte genotype, and found increased paracellular and transcellular permeability when vessels are formed in the presence of mutant astrocytes. This astrocyte-induced phenotype could be mediated by the secretion of detrimental molecules. In fact, previous studies suggested that inflammatory mediators can alter BBB properties[25,97], and our transcriptomic and biochemical analyses revealed that iPSC-derived LRRK2 G2019S astrocytes acquire a pro-inflammatory profile (Figs. 1 and 4, Supplementary Fig. 2). These observations are particularly compelling as patients with PD and animal models of the disease display (1) increased levels of brain and circulating pro-inflammatory cytokines[8,73,98], (2) pathological neuroinflammation featuring astrocyte reactivity[7] and (3) BBB damage[17–19,99], hinting at the possible association between astrocyte-secreted cytokines and pathological BBB. The BBB-chip model enabling the exploration of this complex triad of events, we therefore pursued the investigation of inflammation-BBB relationships, but we noted that abnormal secretion of angiogenic molecules by astrocytes could also play a role in the observed BBB dysfunction. For example, we found altered patterns of VEGF secretion by astrocytes that mirror changes previously documented in the CSF of PD patients[93,100], and in particular those harboring a LRRK2 G2019S mutation[101]. These observations further strengthen the relevance of our human-based BBB model and the central role that astrocyte-related pathological changes may play in PD.

The LRRK2 kinase appears to be largely implicated in regulating immune-related pathological features, even in idiopathic patients, as suggested by studies showing LRRK2 upregulation/activation in immune cells upon exposure to inflammatory insults[102,103], and in neuronal cells over the course of idiopathic PD[36]. Notably, the secretion of IL-6 and IL-8 by immune cells has been reported in studies focusing on the inflammatory response in people with PD. A clinical study found that increased IL-8, MCP-1 and MIP-1-β levels in the serum of people with LRRK2-PD correlated with severe motor and non-motor clinical subtypes[104]. In a study by Cook et al, patient-isolated circulating immune cells secreted increased levels of IL-6 upon immune challenge, and LRRK2 expression was increased in B cells, T cells and CD16+ monocytes[35]. In another report, iPSC-derived monocytes and macrophages harboring the LRRK2 G2019S mutation were stimulated with toll-like receptor agonists, and the investigators observed increased cytokine secretion (e.g. IL-6, IL-8) in the mutant cells compared to isogenic controls[81]. The increased release of IL-6 and IL-8, associated with changes to LRRK2 expression or kinase activity, corroborates our findings related to LRRK2 G2019S astrocytes. An interesting report suggests that LRRK2 does not regulate cytokine release in the same manner in all cell types[105]. In their work, the authors claim that iPSC-derived microglia carrying the LRRK2 G2019S mutation decrease their secretion of IL-6 and IL-8 upon lipopolysaccharide stimulation[105]. While the specific biological consequences of cytokine secretion on PD-related pathological features remains to be further investigated, the present study proposes a link between inflammation and alterations at the BBB using a human model recapitulating the architectural complexity of the NVU.

As we learn more about the roles of individual cells types forming the NVU, inclusion of neurons and microglia in future studies would increase the complexity of our BBB chip. Animal models of PD could also be leveraged to confirm the role of reactive astrocytes on BBB integrity. Under native conditions, cell types forming the NVU may be able to compensate for, at least to some extent, dysfunctional cells including astrocytes[106].

To investigate the implications of astrocyte-derived inflammatory mediators to BBB permeability, we first sought to identify signaling pathways driving the observed astrocyte reactivity. Our data suggest a constitutive activation of ERK1/2 in LRRK2 G2019S astrocytes. This kinase cascade is involved in the regulation of cellular responses to stressors and, together with other kinase cascades, contributes to mounting a cellular response to inflammation[107–109]. The LRRK2 G2019S mutation had previously been shown to promote ERK1/2 phosphorylation[110] and inhibition of the upstream MEK1/2 kinase rescued features of neurodegeneration in iPSC-derived dopaminergic neurons[70]. Consistent with this observation that MEK/ERK signaling is implicated in PD-related pathological alterations, pharmacological inhibition of MEK1/2 kinase activity rescued the inflammatory profile of LRRK2 G2019S astrocytes and attenuated cytokine secretion (Fig. 5). When added to the brain compartment of a BBB^CTL, these cytokines increase vessel permeability, suggesting that reducing astrocyte reactivity may improve this phenotype. Indeed, MEK1/2 inhibition in the brain compartment reduced IL-6 and IL-8 release and rescued vessel formation and integrity, which was concomitant with the activation of the pro-survival AKT signaling pathway (Figs. 6 and 7). However, the MEK/ERK signaling cascade is involved in a large number of biological functions and its inactivation could mitigate other detrimental processes unrelated to inflammation and outside the scope of this study. For example, ERK1/2 was shown to mediate BBB alterations via the phosphorylation of astrocytic connexin43 during brain ischemia, and pharmacological as well as genetic suppression of this signaling cascade preserved BBB integrity[111]. Regardless, the data presented here suggests that MEK1/2 is central to regulating astrocyte paracrine signaling and is a common factor regulating two critical astrocytic functions, i.e. CNS immunity and BBB support. This finding also corroborates a study by

Kim et al, demonstrating that astrocyte reactivity induced by the TNF-STAT3 signaling axis leads to BBB dysfunction[63]. Another important finding is the observation that rescue of BBB$^{G2019S}$ function is associated with AKT phosphorylation and presumably with activation of this protective kinase cascade. Previous studies reported AKT activation as a key molecular mechanism involved in the preservation of BBB integrity by mesencephalic astrocyte-derived neurotrophic factors[82] and in a mouse model of traumatic brain injury[83]. Another report found that Granulocyte-colony stimulating factor (G-CSF), a neuroprotective growth factor, preserved BBB integrity in a mouse model of neonatal hypoxia via AKT activation and downregulation of inflammatory molecules[112]. Our work suggests that the LRRK2 G2019S mutation affects several signaling pathways involved in astrocyte-BMEC communication at the BBB, and elucidating the specific secreted factors mediating these interactions would further advance our understanding of BBB regulation in health and disease. Our data also proposes that astrocytes may be a key cell type that bridges pathological processes occurring in CNS and the periphery via a dysfunctional BBB. Further studies are needed to better elucidate these mechanisms, which could unite the two existing models of CNS and peripheral roots of PD and open new therapeutic avenues to treat patients. In addition, clinical studies correlating the onset of BBB leakage to pathology in the brain and periphery will be critical to determine if loss of barrier integrity is an early or late event in the disease course.

By modeling the complex brain vasculature in vitro, we identified astrocyte-related pathological changes to vessel function and morphology. Postmortem analysis of PD vs. control substantia nigra tissue corroborated our observations of vessel enlargement (Fig. 8). The investigation of postmortem brain samples revealed additional disease-related modifications of the brain vasculature in the substantia nigra of PD patients, as demonstrated by changes to the number and total coverage area of blood vessels. These findings are consistent with an earlier study that suggested decreased blood vessel density and increased vessel diameter in the postmortem brain of PD patients[86], and with an α-synuclein overexpression mouse model that documented BBB leakage associated with lower striatal vessel density compared to wild-type animals[113]. However, the specific role of astrocytes in human vascular pathology should be addressed in future studies, for example by correlating postmortem quantification of vascular changes with astrocyte reactivity and angiogenic potential. In addition, exploring these changes in a larger cohort of patients would enable in-depth statistical analyses that consider the impact of covariates such as comorbidities, treatment regimen, or biological sex.

Collectively, this study proposes a framework where BBB impairments may be, at least in part, mediated by disease-induced astrocyte alterations. The LRRK2 G2019S mutation appears to maintain astrocytes in a reactive state and dysregulate genes involved in angiogenesis, which could in turn affect the growth of functional capillaries. In addition, the results presented here identify MEK1/2 as a key regulator of astrocyte paracrine signaling, and suggest that a shared signaling cascade may mediate both inflammation and loss of barrier integrity in PD. Our findings reinforce previously published studies detailing astrocyte dysfunction in PD and suggesting the alteration of several key molecular mechanisms mediating cell-cell communication[13,14].

## Methods

### Key resources table
See Supplementary Table 2.

### Culture of iPSCs and pericytes
Three independent patient-derived iPSC lines with the PD-related mutation LRRK2 G2019S were used to perform this study (Supplementary Tables 1 and 2). A first pair consisting of LRRK2 G2019S iPSCs derived from female donors and their corresponding gene-edited isogenic controls were kindly provided by Prof. Dr. Thomas Gasser (Universitätsklinikum Tübingen) and Prof. Dr. Hans R. Schöler (Max-Planck Institute)[70]. The second pair, provided by Dr. Randall T. Moon (Howard Hughes Medical Institute/University of Washington), was generated via the reprograming of dermal fibroblasts available through the Coriell Institute for Medical Research via the NINDS repository (ND 33879, female LRRK2 G2019S; ND 36091, female control)[13]. Additional LRRK2 G2019S and control iPSC lines were obtained directly via the NINDS repository (LRRK2 G2019S, ND 40018, female; control, ND 38554, female). Informed consent was obtained from donors as detailed in[70] and in the Coriell Institute for Medical Research guidelines (https://catalog.coriell.org/0/Sections/Collections/NINDS/InvestigatorFAQ.aspx?PgId=150). iPSCs were maintained in mTeSR Plus medium (StemCell Technologies, Vancouver, BC, Canada), passaged as small aggregates using ReLeSR (StemCell Technologies) and plated onto Geltrex-coated culture dishes (Thermo Fisher Scientific, Waltham, MA). Genomic stability was assessed regularly via karyotype analysis (Cell Line Genetics, Madison, WI).

Primary human brain vascular pericytes (ScienCell Research Laboratories, Carlsbad, CA) were cultured in complete pericyte medium (ScienCell Research Laboratories), and 80% confluent monolayers were passaged using a 0.25% Trypsin-EDTA solution for maintenance into TC-treated T25 flasks without pre-coating.

### Preparation and culture of iPSC-derived cells
*BMEC-like cells.* IPSCs used to prepare BMEC-like cells were passaged and maintained in E8 medium (StemCell Technologies), and differentiated as described before[114,115]. IPSCs were incubated in Accutase (Millipore Sigma, Burlington, MA), dissociated into single cells and plated at a density of 12,500 cells/cm² on Geltrex-coated plates in E8 medium supplemented with ROCK inhibitor (Y-27632, StemCell Technologies) (Day −1). The next day, cells were washed with PBS and cultured in E6 medium (StemCell Technologies) with daily media changes for 4 days (Days 0-3). On Days 4 and 5, media was changed to endothelial cell (EC) medium consisting of human endothelial serum-free medium (Thermo Fisher Scientific) supplemented with 1% platelet-poor human plasma (Sigma), 20 ng/mL FGFb (StemCell Technologies) and 10 μM retinoic acid (Sigma). On Day 6, the cells were harvested in Accutase and plated on Geltrex-coated dishes or in BBB microfluidic chips for experiments. After replating, BMEC-like cells were cultured in EC⁺ (EC medium supplemented with 20 μM RO-20-1724, 400 μM dibutyryl cAMP and 10 μM retinoic acid), and 10 μM ROCK inhibitor was added for the first 24 to 48 h to improve survival as described before[116]. BMEC-like cells were newly differentiated before each BBB-chip experiment.

*Astrocytes.* IPSCs used to prepare astrocytes were passaged and maintained in mTeSR Plus medium, and differentiated as described before[13,55]. Briefly, iPSCs were first neuralized by dual SMAD inhibition[117] to generate midbrain-patterned neural progenitor cells (NPCs), which were subsequently differentiated into astrocytes by culturing and subpassaging cells with astrocyte medium (ScienCell Research Laboratories). The newly differentiated astrocytes were cryopreserved for later use or plated for experiments. For monolayer cultures, astrocytes were plated at 40,000 cells/cm² and used to extract protein lysates, isolate RNA or harvest ACM. As needed, cultures were treated with PD0325901 (0.5 μM, Cayman Chemical Company, Ann Arbor, MI) or SCH772984 (0.5 μM, Cayman Chemical Company) for 24 h. Overall, up to four independent astrocyte differentiations were performed for each iPSC lines, and each batch produced a large number of cryopreserved aliquots for long term storage and experimental use.

## RNA-sequencing meta-analysis

RNA-seq datasets were obtained from GEO repository using accession numbers GSE116124[14], GSE152768[13], GSE120306[27]. Reads were trimmed using fastp v0.20.0[118]. Quality check was performed on raw and trimmed data to ensure the quality of the reads using FastQC v0.11.8[119] and MultiQC v1.8[120]. The quantification was performed with Kallisto v0.46.2[121] against the *Homo sapiens* transcriptome (downloaded from Ensembl release 100). Principal component analysis was completed with the FactoMineR v2.4[122] and ggplot2 v3.3.3[123] R packages. Raw counts were normalized using RUVSeq v1.24.0[124] with the housekeeping genes from Jongue et al.[125]. Differential expression analysis was performed using the DESeq2 v1.30.1 package[126]. All R analyses were done in R v4.0.3[127]. Gene ontology enrichment analysis was performed using the Database for Annotation, Visualization, and Integrated Discovery (DAVID v6.8, https://david.ncifcrf.gov/)[128,129]. Histograms provided in Fig. 1 include genes that were found to be differentially regulated in at least two datasets. To calculate the percentage of genes from the Angiogenesis, Inflammation and Cell Adhesion pathways that are targeted by an ERK-related transcription factor (TF), we first extracted all the TF-genes interactions from the Human Transcription Factors database[130], the HumanTFDB portal[131] and the TF2DNA database[132]. Genes from the ERK pathway were then defined as such if they were detected in at least one of the following database: QuickGO[133] (GO:0070371: ERK1 and ERK2 cascade), GSEA[134] (ST_ERK1_ERK2_MAPK_PATHWAY), Biocarta[135] (erk1/erk2 mapk signaling pathway) or KEGG[136] (map04010: MAPK signaling pathway). The percentage of genes targeted by ERK TF in each of the previously defined signaling pathways was then calculated using the TF-gene interactions filtered to keep only the ERK TF.

### GSEA analysis.

GSEA analysis was performed using the broad institute GSEA-P tools[134] and a modified MSigDB curated gene sets version 5.1 where "VART_KSHV_INFECTION_ANGIOGENIC_MARKERS_DN" and "VART_KSHV_INFECTION_ANGIOGENIC_MARKERS_UP" were merged into a single entry. The heatmap of the presence/absence of core enrichment gene across pathways of interest was produced from the GSEA results using a custom R[127] script and the tidyverse v2.0.0[137] and the ComplexHeatmap v2.6.2[138] packages.

## 3D microfluidic BBB

OrganoPlate 3-lane 40 (Mimetas, Gaithersburg, MD) were used to prepare 3D models of the BBB on individual tissue culture chips. Each chip consists of a top lane used to grow the endothelial-like vessel, a middle lane filled with an ECM gel and a bottom lane plated with human primary pericytes and iPSC-derived astrocytes.

### Coating procedure.

The day prior to plating, the middle perfusion lane was injected with 1.8 µl of a collagen I solution (7.5 mg/mL, Corning), briefly incubated at 37 °C for 10 to 15 min before adding 50 µl HBSS to prevent dehydration, and allowed to further solidify for 1 h at 37 °C. The top perfusion channel was subsequently injected with 1.8 µl of a fibronectin (50 µg/mL, Sigma-Aldrich) and collagen IV (330 µg/mL, Sigma-Aldrich) coating solution and returned to the cell culture incubator until plating.

### Cell plating.

IPSC-derived BMEC-like cells were washed once with Dulbecco's phosphate-buffered saline (DPBS) and dislodged in Accutase for 20 min at 37 °C. The cells were then centrifuged at 180 x g for 5 min and resuspended in EC$^+$ medium supplemented with 10 µM ROCK inhibitor. Human primary pericytes were washed once with DPBS, incubated in 0.25% trypsin/EDTA for 1 min, centrifuged at 180 × $g$ for 5 min and resuspended in astrocyte medium. Cryopreserved iPSC-derived astrocytes were thawed at the time of plating, or maintenance cultures were harvested in Accutase for 5 min. Cells were then centrifuged at 180 × $g$ for 5 min and resuspended in astrocyte medium.

A 7 × 10$^4$ BMEC-like cell/µL suspension was prepared and 2 µl were injected in the top left inlet using an electronic single channel pipette. A mixed cell suspension consisting of 7.5 × 10$^3$ astrocytes/ µL and 7.5 × 10$^3$ pericytes/ µL was then prepared and 2 µl were injected in the bottom left inlet. A total of 50 µl of EC$^+$ medium supplemented with 10 µM ROCK inhibitor was added to the top left inlet, and 50 µl of astrocyte medium was added to the bottom left inlet. The plate was placed in an upright position using a plate holder provided by the manufacturer and placed in the cell culture incubator. The cells were allowed to attach for 3 to 4 h at 37 °C before replenishing all wells with astrocyte medium supplemented with PD0325901 (0.5 µM, Cayman Chemical Company), recombinant human IL-6 or IL-8 (100 ng/mL) (PeproTech, Cranbury, NJ) as needed in the brain compartment, and EC$^+$ with ROCK inhibitor in the vascular compartment. The plate was then placed on a perfusion rocking platform (Mimetas) set at 0.1 cycles/min. After 48 h in culture, EC$^+$, astrocyte medium and drug treatments were replenished, and the perfusion rocker speed was increased to 0.2 cycles/min.

## BBB-chip barrier integrity assay

After 4 and 6 days in culture, media in the ECM and brain compartments was replaced with fresh EC medium (20 µl in the middle and bottom inlets and outlets), and the vascular compartment was replenished with fresh EC medium supplemented with fluorescein (500 µg/mL, Thermo Fisher Scientific), dextran-TMRE (500 µg/mL, Millipore Sigma) or rhodamine B (250 µg/mL, Millipore Sigma) (40 µl in the top left inlet and 30 µl in the top right outlet). The plate was imaged immediately using an EVOS cell imaging system (Thermo Fisher Scientific) equipped with a ×4 objective, and a total of 6 images were obtained at an interval of at least 6 min between each time-lapse acquisition. The field of view in the microscope was set up so that the three compartments (vascular, ECM and brain) were visible in the images. Fluorescence intensity representing dye migration from the vascular to the brain compartments was quantified using ImageJ (version 1.53 T)[139] and P$_{app}$ values were calculated using the following formula:

$$P_{app}\_value(cm/s) = (I\_end - I\_initial)/(T\_end - T\_initial) * V\_gel/(A\_barrier),$$

where cm = centimeter, s = seconds; I_intial = initial intensity; I_end = endpoint intensity; T_initial = time intial in seconds; T_end = time end in seconds; V_gel = 1.04 × 10$^4$, and A_barrier = 5.7 × 10$^3$. A_Barrier and V_gel were considered constants since the area available for molecule exchange was constrained by the dimensions of the plate. In these conditions, the assumption is that the fluid exchange area between the vessel and the brain compartments remains constant regardless of vessel width.

## Transwell barrier integrity assay

BMEC-like cells were differentiated from control iPSCs as described above. On Day 6, the cells were harvested in Accutase and plated on Geltrex-coated 6.5 mm transwell inserts (Millipore Sigma) at a density of 35,000 cells/insert in 100 µl EC$^+$ supplemented with 10 µM ROCK inhibitor. Human primary pericytes, together with control or LRRK2 G2019S astrocytes, were added to the bottom Geltrex-coated chamber of the transwells, at a density of 60,000 pericytes and 60,000 astrocytes per well, in 600 µl astrocyte medium. The next day, the EC+ medium was removed and replaced with EC+ supplemented with 500 µg/mL 4.4 kDa dextran-TMRE for 12 h. Media was subsequently collected from the top and bottom chambers, and fluorescence was quantified using a microplate reader (BioTek Synergy).

## Immunofluorescence

**Monolayers.** Cells grown on Geltrex-coated German glass coverslips (Electron Microscopy Sciences, Hatfield, PA) were washed once with DPBS and fixed in 4% paraformaldehyde (Fisher Scientific, Waltham, MA) for 20 min at room temperature. The cells were then washed once

with DPBS to remove traces of paraformaldehyde, blocked/permeabilized in blocking buffer (0.3% TX-100, 1% BSA, 10% FBS in DPBS) for 1 h at room temperature and incubated in primary antibody overnight at 4 °C with the following antibodies: anti-GFAP (1:500, BD Biosciences, Franklin Lakes, NJ), anti-ZO-1 (1:500, Invitrogen, Waltham, MA), anti-VE-cadherin (1:500, R&D), and anti-claudin-5 (1:500, Thermo Fisher Scientific) in BSA buffer (1% BSA in PBS). Alternatively, cultures were not permeabilized but directly incubated in blocking buffer (1% BSA, 10% FBS in DPBS) for 1 h at room temperature, followed by anti-occludin (1:200, Invitrogen) primary antibody overnight at 4 °C. The next day, the cells were washed twice with DPBS, then incubated in Alexa-conjugated secondary antibodies of the appropriate species (goat anti-mouse, anti-rabbit, donkey anti-rabbit, or donkey anti-goat; Alexa Fluor 488, or Alexa Fluor 555; Thermo Fisher Scientific) diluted 1:1000 in BSA buffer for 1 h at room temperature. Lastly, the cells were washed three times with DPBS prior to mounting coverslips on slides with ProLong Diamond with DAPI (Thermo Fisher Scientific) and cured for 24 h in the dark at room temperature.

**BBB microfluidic plate.** Cultures in the microchips were washed once with DPBS and fixed in 4% paraformaldehyde (Fisher Scientific) for 20 min at room temperature. The chips were then washed once with DPBS, blocked/permeabilized in saponin buffer (0.1% saponin, 1% BSA, 10% FBS in DPBS) for 1 h at room temperature, and incubated overnight at 4 °C with primary antibodies (1:50 dilution) in saponin buffer without FBS. The following antibodies were used: anti-GFAP (BD Biosciences), anti-Alpha-Smooth Muscle Actin (aSMA) (Sigma), anti-Glut1 (Novus Biologicals, Toronto, Canada), anti-ZO-1 (Invitrogen). The next day, the cells were washed twice with DPBS, then incubated in Alexa-conjugated secondary antibodies of the appropriate species (goat-anti rabbit, goat-anti mouse, donkey anti-mouse; Alexa Fluor 488; Alexa Fluor 555; Thermo Fisher Scientific) and DAPI nuclear stain diluted 1:500 in saponin buffer without FBS for 1 h at room temperature. Lastly, the cells were washed three times with DPBS and stored at 4 °C.

## Immunocytochemistry of human postmortem sections

Ethical approval for the use of human postmortem tissue was received from the Institutional Review Board (Comité d'éthique de la recherche) overseeing research performed at the Centre de Recherche du CHU de Quebec-Université Laval (approval #A13-02-1138). Brain tissue was obtained from the CERVO Brain Bank (Quebec City, Canada), and written informed consent was obtained before tissue donation (ethical approval provided by the Institutional Review Board CIUSSS de la Capitale-Nationale, projects #2013-3 and #146). Donors consented to the inclusion of indirect identifiers, such as age, sex, or cause of death, in published studies. The information about biological sex was obtained from the patients' medical records.

Free floating sections of the substantia nigra were incubated in 4% PFA pH 7.4 for 1 h, washed three times with potassium phosphate buffer saline (KPBS; 22 mM potassium phosphate dibasic anhydrous, 3.3 mM potassium phosphate monobasic anhydrous, 53 mM sodium chloride) and incubated in Trilogy™ pretreatment solution for 30 min at 95 °C, followed by a cool down at room temperature in the same solution for 20 min. The sections were then washed three times with KPBS, and incubated in blocking buffer (10% (v/v) donkey serum, 0.1% (v/v) Triton-X, and 0.5% (w/v) BSA in KPBS) for 30 min. The sections were subsequently incubated overnight at 4 °C in anti-laminin (1:250 dilution, Agilent Dako, CA) and anti-tyrosine hydroxylase (1:500 dilution, Millipore Sigma) primary antibody prepared in blocking buffer. The next day, the samples were washed three times with KPBS and incubated in Alexa-conjugated secondary antibodies (goat anti-mouse Alexa 488, donkey anti-rabbit 546; 1:500 to 1:250 dilution, Thermo Fisher Scientific) prepared in blocking buffer for 2 h at room temperature, followed by two washes with KPBS and a 7 min incubation in KPBS buffer supplemented with DAPI nuclear stain. The sections were

washed twice in KPBS buffer and mounted on slides. Once dried the slides were incubated in 70% ethanol for 5 min, followed by Sudan black for another 5 min, and three ethanol washes. The sections were dried and embedded in Fluoromount-G mounting medium (Invitrogen) for long-term storage and visualization.

## RNA isolation and RT-qPCR

Cells were grown as confluent monolayers, washed with DPBS and homogenized in TRIzol (Thermo Fisher Scientific) for immediate processing or storage at −80 °C. Samples were vigorously mixed with chloroform (20% v/v) and centrifuged at 12,000 × g for 15 min, at 4 °C, to separate the aqueous solution from the organic phase. The aqueous phase was collected, mixed 1:1 with ethanol and processed using an RNeasy kit (Qiagen, Hilden, Germany) or a Direct-zol RNA Miniprep kit (Zymo Research, Irvine, CA) following the manufacturer's instructions. Isolated RNA was stored at −80 °C or processed immediately for cDNA synthesis using a RevertAid First Strand cDNA Synthesis Kit (Thermo Fisher Scientific) as described in the manufacturer's instructions. Forward and reverse primers for RT-qPCR were designed at exon-exon junctions using Beacon Designer Lite 8.16 (Premier Biosoft, San Francisco, CA), and the primer sequences are available in Supplementary Table 3. RT-qPCR reactions were performed using the KAPA SYBR FAST master mix (Roche, Basel, Switzerland) and a Roche LightCycler 480 or 96 System. Relative mRNA levels were calculated for each gene using the formula:

$$2^{-\Delta\Delta Ct} = 2^{-((Ct,TG-Ct,\beta actin)\,LRRK2\,G2019S-(Ct,TG-Ct,\beta actin)\,control)}$$

where "Ct, TG" represents the cycle threshold (Ct) for the target gene (TG), and "Ct, βactin" represents the cycle threshold for the loading reference *ACTB* (β-actin).

## Confocal microscopy

The following microscopes were used: (1) Zeiss LSM 900 inverted laser scanning confocal microscope equipped with a Plan-Apochromat 10× objective lens (Zeiss, NA = 0.45), a Plan-Apochromat 20× objective lens (Zeiss, NA = 0.8) and 405 nm, 488 nm, 561 nm and 640 nm laser lines; (2) Zeiss LSM 880 inverted laser scanning confocal microscope equipped with a 40× oil objective lens (Zeiss, NA = 1.3), 405 nm, 488 nm and 561 nm laser lines, and ZEN software version 2.3 (Zeiss, Oberkochen, Germany).

## Western blot

**Monolayers.** Cell grown in monolayers were extracted using 1X RIPA buffer (Cell Signaling Technologies, Danvers, MA) supplemented with 1X Halt protease and phosphatase inhibitor cocktail (Thermo Fisher Scientific), and lysates were centrifuged at 15,000 × g at 4 °C for 15 min. Supernatants were collected, protein concentration was quantified using a BCA assay (Thermo Fisher Scientific) and samples were mixed with sample buffer before heating at 65 °C for 15 min. Lysates were loaded onto a 4-20% Tris-Glycine gel (ThermoFisher Scientific), protein migration was performed using a Tris-Glycine SDS running buffer (Thermo Fisher Scientific) and proteins were transferred using a Trans-Blot Turbo system (Bio-Rad) set to the "mixed molecular weight" setting. Membranes were subsequently blocked with a 4% BSA in PBST (PBS supplemented with 0.1% Tween 20) buffer for 1 h at RT, and incubated with the following primary antibodies at 4 °C overnight: anti-p44/42 MAPK (1:2000, Cell Signaling Technologies), anti-phospho-p44/42 MAPK (Thr202/Tyr204) (1:3000, Cell Signaling Technologies) and anti-GAPDH (1:5000, Cell Signaling Technologies). The next day, membranes were washed three times in PBST and incubated with an HRP-conjugated secondary antibody (1:25,000 dilution in blocking buffer) of the appropriate species for 1 h at RT. Membranes were washed three times in PBST and revealed using a Clarity Western ECL Substrate (Bio-Rad Laboratories, Hercules, CA) on a ChemiDoc MP

imaging system (Bio-Rad). Quantification of protein levels was performed using ImageJ (version 1.51n)[139].

**Microfluidic chips and BMEC-like cell monolayers.** Vessels grown in 2 to 3 chips were pooled and extracted in 30 µl RIPA buffer (150 mM NaCl, 5 mM EDTA pH 8, 50 mM Tris pH 8, 1% (v/v) NP-40, 0.5% (w/v) sodium deoxycholate, 0.1% (w/v) SDS in water) supplemented with 1X Halt protease and phosphatase inhibitor cocktail (Thermo Fisher Scientific) on ice. Protein lysates were heated at 95 °C for 5 min, loaded onto an 8% SDS-PAGE gel and proteins were subsequently blotted onto a 0.45 µm pore size hydrophobic polyvinylidene fluoride (PVDF) transfer membrane (Thermo Fisher). Protein transfer was carried out overnight at 20 V, and the buffer consisted of 0.3% (w/v) Tris-Base, 1.5% (w/v) glycine and 10% methanol (v/v) in deionized water. Membranes were incubated in blocking buffer (2.5% (w/v) BSA in PBS supplemented with 0.1% (v/v) Tween20) for 1 h at RT, and incubated with the following antibodies at 4 °C overnight: anti-ZO-1 (1:500, Thermo Fisher Scientific), Anti-VE-cadherin (1:1000, Abcam, Cambridge, UK), anti-claudin-5 (1:500, Thermo Fisher Scientific), anti-phospho-AKT (Ser473) (1:1000, Cell Signaling Technologies), anti-AKT1/2/3 (1:1000, Santa Cruz Biotechnology), anti-p38 MAPK (1:1000, Cell Signaling Technologies), anti-phospho-p38 MAPK (Thr180/Tyr182) (1:1000, Cell Signaling Technologies), and anti-GAPDH (1:5000, Applied Biological Materials, Richmond, BC, Canada) diluted in SuperBlock with 0.1% (v/v) Tween20 (Thermo Fisher Scientific) or in blocking buffer (2.5% (w/v) BSA in PBS). The next day, membranes were washed three times with PBS 0.1% (v/v) Tween20, incubated with corresponding HRP-conjugated secondary antibodies from appropriate host species (1:25,000 dilution in blocking buffer) for 1 h at RT and washed three times before revelation using ECL substrate (Immobilon Forte Western HRP substrate, Millipore) on a MyECL Imager system (ThermoFisher Scientific). Quantification of protein levels was performed using myImageAnalysis Software (Thermo Fisher Scientific, version 1.1).

**Proteome profiler arrays**
**Angiogenesis membrane array.** Conditioned medium was prepared from 80 to 90% confluent astrocyte cultures and cell debris were removed by centrifugation at 2000 × g for 10 min. A Proteome Profiler Human Angiogenesis Array Kit (R&D Systems, Minneapolis, MN) was incubated with control or LRRK2 G2019S ACM following the manufacturer's guidelines. Membranes were revealed using a Clarity Max Western ECL Substrate (Bio-Rad Laboratories) on MyECL Imager system (ThermoFisher Scientific), and quantified using myImageAnalysis Software (Thermo Fisher Scientific, version 1.1).

**ELISA**
Conditioned medium was prepared from 80 to 90% confluent astrocyte cultures or collected from the BBB microfluidic plates and either sterile filtered or centrifuged at 2000 × g for 10 min to remove cell debris. Samples were frozen as single-use aliquots at −20 °C and thawed on ice before performing cytokine measurement using the following ELISA kits according to the manufacturer's instructions: IL-6 (BBB microfluidic plate: R&D Systems; monolayer cultures: Thermo Fisher Scientific), IL-8 (*BBB* microfluidic plate: R&D Systems; monolayer cultures: Thermo Fisher Scientific).

**Statistical analysis**
Data was analyzed using GraphPad Prism version 8.0 (La Jolla, CA). P-values for experiments consisting of two groups were calculated using two-tailed unpaired Student's *t* test assuming equal standard deviation, or a one-sample t-test to compare the mean of each sample to a hypothetical mean of 0 (log-corrected values) or 1 (uncorrected values). Experiments with multiple groups were analyzed via one-way ANOVA with Dunnett's multiple comparisons correction, or two-way ANOVA with Šídák's multiple comparisons test. Outliers in were

identified using Grubbs' test with an alpha value set at 0.05 and removed from analysis.

**Reporting summary**
Further information on research design is available in the Nature Portfolio Reporting Summary linked to this article.

## Data availability
RNA-seq datasets analyzed in this study were previously published and are publicly available, and they were obtained from the GEO repository using accession numbers GSE116124, GSE152768, GSE120306. This study also accessed the following databases: Ensembl release 100 (https://www.ensembl.org/info/website/archives/index.html?redirect=no), the Human Transcription Factors database (http://humantfs.ccbr.utoronto.ca/), the HumanTFDB portal (http://bioinfo.life.hust.edu.cn/AnimalTFDB/), the TF2DNA database (https://www.fiserlab.org/tf2dna_db/), QuickGO (https://www.ebi.ac.uk/QuickGO/), GSEA (https://www.gsea-msigdb.org/gsea/index.jsp), Biocarta (https://maayanlab.cloud/Harmonizome/dataset/Biocarta+Pathways), KEGG (https://www.genome.jp/kegg/). Source data are provided with this paper.

## Code availability
Custom code generated to analyze the RNA-seq data is available at: https://github.com/ArnaudDroitLab/de_rus_jacquet_2023.

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

## Acknowledgements

The authors would like to thank members of the Cicchetti lab for helpful discussions and feedback. We warmly thank Dr. Erin K. O'Shea for supporting the work performed by A.d.R.J. in her laboratory at Janelia research campus. We thank Prof. Dr. Thomas Gasser

(Universitätsklinikum Tübingen) and Prof. Dr. Hans R. Schöler (Max-Planck Institute) for providing the iPSCs with the LRRK2 G2019S mutation and isogenic control, and Dr. Randall T. Moon (Howard Hughes Medical Institute/University of Washington), for providing the non-isogenic iPSC lines. We would also like to thank Ms. Christine Légaré for assistance with the use of the Zeiss LSM900 microscope, and Ms. Marie-Josée Wallman for assisting with the preparation of human postmortem brain sections. A.d.R.J. was supported by a post-doctoral fellowship and a Launch Award from the Parkinson's Foundation (PF-PRF-835671, PF-Launch-937655). M.A. was supported by postdoctoral fellowships from the Canadian Institutes of Health Research (CIHR, 201910MFE-430883-231775) and from the *Fonds de Recherche du Québec en Santé* (FRQS, 260387). H.L.D. was supported by a doctoral research award from FRQS (award number 272842). Work performed in the laboratory of E.K.O at Janelia Research Campus was supported by the Howard Hughes Medical Institute. F.C. is a recipient of a Researcher Chair from FRQS providing salary support and oper-ating funds (award number 35059). The Zeiss LSM900 microscope was acquired using a John R. Evans Leaders fund from the Canada Foundation of Innovation (S.B.). S.B. is the recipient of the Canadian Research Chair in Epithelial Dynamics of the Kidney and Reproductive Organs. A.D. was supported by L'Oréal Research Chair in Digital Biol-ogy. Author contributions follow the CRediT taxonomy available at https://casrai.org/credit.

## Author contributions

A.d.R.J.: Conceptualization, Investigation, Formal Analysis, Metho-dology, Validation, Supervision, Visualization, Project administra-tion, Writing—original draft. M.A.: Investigation, Formal Analysis, Methodology, Validation, Visualization, Writing—review & editing. H.L.D.: Investigation, Formal analysis, Visualization, Writing—Review & Editing. J.L.T.: Investigation, Writing—Review & Editing. M.B.: Investigation. J.D.: Data curation, Software. C.B.: Data curation, Software. L.H.: Data curation, Software. M.S.-P.: Investigation. M.P.: Resources (postmortem brain tissue), Supervision. A.D.: Data cura-tion, Software, Supervision. S.B.: Resources (Zeiss LSM900 micro-scope), Writing—Review & Editing. F.C.: Funding acquisition, Supervision, Writing—Review & Editing.

## Competing interests

The authors declare no competing interests. J.L.T. contributed to this article before her affiliation with the Cell Biology R&D at Thermo Fisher Scientific.
