## [Peer Review File · Nature Communications]

The contribution of inflammatory astrocytes to BBB impairments in a brain-chip model of Parkinson's diseaseReviewers' comments:

Reviewer #1 (Remarks to the Author):

In this work de Rus Jacquet et al examined the role of the LRRK2 Parkinson's disease (PD) associated mutation G2019S in determining proinflammatory changes that affect the integrity of the blood brain barrier (BBB). To this end, they employed patient induced pluripotent stem cells (iPSCs) and established a microfluidic 3D human BBB-chip. Using this device, the authors show that LRRK2 G2019S leads to inflammatory changes in astrocytes and defects in the formation of a functional BBB in vitro.

This work is relevant for the field as it shows that human iPSC organ-on-a-chip models can be employed to investigate BBB dysfunction in the context of human disease. However, the i) relevance and contribution of the current data to our further understanding of LRRK2-PD related mechanisms, and ii) the novelty of the chip should be better addressed. The major issues are as follows:

Major Issues

- The novelty and the advantages of the chip over existing models is unclear and should be better addressed (PMID: 31173718; PMID: 34625559). The authors claim "The distinct advantage of this platform over other existing 3D models is the replacement of a membrane physically separating the vascular and brain compartments with an ECM scaffold that supports vessel formation, cell migration and free diffusion of secreted factors". However, in the current study the authors do not investigate cell migration within their chip model. Furthermore, the chip described by Vatine et al contained a porous membrane and allowed the investigation of diffusion of factors and their impact on iPSC-neurons that were co-cultured in the brain channel (PMID: 31173718). Indeed, the addition of neurons within the chip described by de Rus Jacquet et al would have further strengthened the relevance of the described mechanisms in LRRK2-PD neurodegeneration.
- The study is also limited by the fact that key experiments have been performed using only one isogenic cell couple without further validation (i.e. Protein array in Figure 2I). Given the high variability between the two G2019S mutant lines (see for instance cytokine production in Figure 4 and differentiation yield), additional lines should have been included.
- As described below, the authors performed a meta-analysis that shows the dysregulation of specific genes and pathways in LRRK2 mutant astrocytes. However, none of these candidates is further investigated in subsequent functional analysis. Similarly, the protein array performed in isogenic LRRK2 iPSC-astrocytes show very minor differences. The graph in 1J shows significant differences only for 3 candidate proteins. Based on the data, how can the authors conclude that the G2019S mutation is linked to angiogenesis-related changes?
- The comparison of the PD findings to AD APOE and AD PSENΔE9 results is interesting. However, as the dysfunction of BBB has emerged as a potential pathogenetic factor in AD, these results are quite surprising. To conclude that their findings are rather specific to PD, it would be valuable to consider adding additional iPSC findings with genetic alterations associated with AD. This will be particularly important in consideration of the high variability among different lines and differentiation protocols (overall their findings seem to be largely driven by the study by Di Domenico et al.). This issue should be further discussed; are different mechanisms (genotype specific) involved in BBB dysfunction in PD vs AD? Is there evidence of an increased BBB disruption in LRRK2-PD patients?
- Figure 6D: regarding the image with G2019S astrocytes, one may wonder whether this is very representative of their findings. The highlighted white area shows the failure of vessel formation; however, previous images do not indicate such a significant change. Also, Western blot results (Figure 3E) show no differences in any of the BBB markers that could explain such a change. With this regard, the authors link the BBB dysfunction to the ERK pathway and performed rescue experiments measuring the VE-cadherin, claudin 5, ZO-1 levels. However, in Figure 3E, none of these shows significant changes in LRRK2-G2019S astrocytes.

Additional comments:

1. Figure 1 (Meta-analysis of RNA sequencing data sets): the authors should validate the DEG

emerged from the metanalysis and relevant to the angiogenesis/immune dysregulation pathways in the isogenic LRRK2 astrocytes employed in the current study.

2. Figure 1B: the authors should show the list of DE expressed genes used for their analyses. Based on the graph and the text, they performed GO analyses based on 21 genes and split them based on up-and down-regulated. It is unclear how many genes went into the analyses? Also, the number seems quite low for GO analysis.

3. Figure 1I, J: we understand that the protein array experiment has been performed on one isogenic iPSC couple. To strengthen the validity of these findings, results from additional lines should be provided. Validation with Western Blot/Elisa should be provided.

4. Supplementary Figure 1: the authors should improve the characterization of the iPSC-derived astrocytes including differentiation yield, inflammatory profile (relevant to the current study), and representative images for all cell lines.

5. Figure 2C: the authors claim a 22-fold increase in VE-cadherin in BMEC-like cells; however, to have a better comparison, they should provide the Western blot for hBMEC (as they did for the gene expression in Figure 2B). The authors state "All samples were run on the same gel, and the bands displayed under the graphs are from the same membrane"; however, they only show individual cropped bands. Also, the variability seems to be extremely high and makes wonder about the robustness of the protocol.

6. Figure 2C: overall, it is difficult to assess the efficiency of differentiation protocol and the validity of the chip model. The stainings for VE-cadherin and Occludin display an unexpected nuclear pattern. Furthermore, the authors should further characterize the vascular channel with additional markers (e.g. claudin 1, CD31). Markers as the BBB glucose transporter GLUT-1 should also be assessed. Similarly, the differentiation of iPSCs into brain microvascular endothelial-like cells should be further characterized.

7. Figure 4: the authors investigate the reactive phenotype of G2019S astrocytes. Did this phenotype also emerge from their metanalysis?

8. The authors identified a significant difference in IL-6 and IL-8 secreted by G2019S astrocytes. Besides their role in neurovascular function, is there any specific known association between these cytokines and G2019S-PD?

9. Suddenly, the authors jump to the analysis of the MEK/ERK pathway, which however did not emerge as one of the candidates in their metanalysis nor in the protein array investigation. It would have been more congruent to focus on one of the newly identified candidates.

10. In the discussion, the authors refer to inflammation, infiltration of peripheral immune cells and impaired BBB maintenance in human tissue but they don't really discuss findings from the primary literature. The study would acquire much more impact if complemented with validation results from post-mortem tissue of LRRK2 carriers. At least, it would be important to have a more detailed discussion about their iPSC findings compared to results from patient tissues samples.

11. Given the ongoing clinical trials with LRRK2 kinase inhibitors, it would be valuable to show whether the mechanisms described here are LRRK2 kinase dependent. This would also strengthen the validity of the model for drug testing.

12. The authors should show levels of LRRK2 for all the iPSC lines astrocytes.

Minor issues:

- Figure 2B: It looks like there is a typo as the heatmap labelling shows CHD5 and not CDH5.

Reviewer #2 (Remarks to the Author):

In this work Rus Jacquet et al seek to interrogate the role of PD astrocyte induced neuroinflammation on the integrity of the BBB using an iPSC derived model and microfluidic device. Whether BBB integrity is impaired in PD remains controversial (Fujita, K. et al. Blood-brain barrier permeability in Parkinson's disease patients with and without dyskinesia. *J Neurol* 268, 2246-2255) and early in disease there is not as much evidence of neuroinflammation as seen in other neurodegenerative diseases such as ALS and AD although there is inflammation late in disease. Thus, the rationale for their study is not quite as strong as the authors suggest and they need to cite a wider range of papers both for and

against the ideas they present (or as challenges for the study).

In general, this work is hindered by the relatively few iPSC lines used in these studies and a poor characterization of the iPSC derived BBB endothelium in addition to non appropriate statistical tests (t-test when should be ANOVA). Transwell studies to show increased permeability with the LRRK2 astrocytes and TEER measurements of the iPSC derived BBB endothelial cells would help to address some of the basic characterization concerns about the iPSC derived BBB and interplay with PD astrocytes. Other major concerns include the integrity of the vascular cells in the microfluidic devices as well as the important positive control of direct inflammation challenge of the control chip barrier properties with LPS (or similar inflammatory cytokine) to establish if the impaired barrier properties are indeed directly related to inflammation and not some other mechanism. Additionally, several studies lack sufficient data for convincing statistical analysis, and the overall clarity of how many independent differentiations, experiments is lacking.

Major Concerns:

1. Given that the initial part of this study focuses on comparing previously published data sets, this work seems to be missing the important comparison of if all 3 differentiations made a similar product especially with the wildly different altered numbers of differentially expressed genes. Even though this is previously published data, information on the patient and control lines needs to be provided. Number of individual lines/clones in each study, Age of onset, age of iPSC generation, source tissue ect. Are the controls isogenic? Also need this data for figure 11. It is clear now that there will always be DEGS when comparing diverse data sets like these – but validation against other data sets is the key aspect. As extracellular matrix tends to be the most variable in these studies – these genes also tend to be the most different between groups but on validation this often falls apart.
2. The number of iPSC lines and clones is again a concern. It is It would also be clearer if the LRRK2 astrocytes were compared to an isogenic corrected control for each patient with at data from at least 3 patients presented throughout these studies.
3. Data presented in Figures 2B, 2G, 3A, 3C, 4E, 5B, 5F, and S2A contain only two biological replicates and are likely not sufficiently powered to perform tests of significance in all cases. Additionally, much of the data presented does not appear to be representative of independent experiments which makes it difficult to assess the reproducibility of the model or of the results generated.
4. The barrier integrity assays presented in Figure 2F-G show high variability between biological replicates and an inability to distinguish any difference in permeability between Fluorescein, 4.4kDa-Dex, and Rhodamine. It is unclear if the high variability is due to the method of assessing permeability (Image J quantification of fluoresce intensity changes in EVOS images) or due to high variability inherent in the model. Additional permeability assessments should be made with various sized dextrans and/or with compounds with known BBB permeabilities to determine the reproducibility and robustness of the barrier and method of assessment. A bigger question is - is it that an effective barrier fails to form or that the formation is delayed in the absence of the astrocytes? Only 1 timepoint at 6 DIV was investigated. Since the BMECs were newly generated how was each set of cells characterized prior to seeding in the chips? This is especially important given the variation in tight junction proteins in 2C. It is also unclear how many independent differentiations are represented in any of the figures
5. In Figure 3 B,C the use of a 2 way ANOVA is misleading. Since each fluorescent dye is examining a separate aspect of the barrier property, these should be treated as separate studies. The appropriate comparison for each dye would be a one-way ANOVA with a Tukey or Sidak multiple comparison test. It is also confusing as to why several conditions only have 2 biological replicates shown (ie 2B control #1 vs G2019S #1) and how this statistical comparison was performed. Furthermore, the authors observe increased permeability in BBB-chips co-cultured with LRRK2 G2019S astrocytes, but no change in tight junction proteins. However, they do note increases in

BMEC-like cell size in BBB-G2019S chips (Figure 3I-K). If similar numbers of BMEC-like cells are in Control and G2019S chips, the larger cells likely result in larger vessels with more surface area. Judging by the shape and size of vessels shown from above in Figure 3A and in cross section in Figure 3D, there does seem to be some differences in vessel area between Control and G2019S. Have the authors accounted for differences in vessel surface area in their permeability measurements (or shown that there is no difference in vessel surface area) due to the increased cell size seen in BMEC-like cells in G2019S chips? If not, differences in area can lead to differences in permeability that could be misinterpreted as differences in barrier.

6. The authors conclude on Line 339 that “formation of leaktight vessels requires the presence of both pericytes and astrocytes (Figure 2F-H)”. However, the data only supports the requirement of astrocytes in the formation of leak-tight vessels. No “Astrocyte Only” condition has been included in Figure 2G-H to determine the necessity of pericytes in the model. It is unclear if the pericytes are required for vessel formation or if they are involved in any of the crosstalk between endothelial cells and astrocytes in the model.

7. The gene expression changes presented in Figure 4A and 4B, which are used to support the claim that G2019S mutation is associated with upregulation of A1, A2, and pan-reactive astrocyte genes, show almost no overlap and often conflicting direction of gene expression change in the G2019S vs. Control comparison across the two sets of lines tested. A third set of G2019S vs healthy control cells should be included, or replication of gene expression changes from independent experiments should be included to determine if the expression changes are consistent or just highly variable from differentiation to differentiation.

8. Given that the ERK pathway is central to proliferation, it is unclear how the total number of cells is controlled for in the figure 5 ERK inhibitor studies? It seems with both the gene expression and ELISA data are only normalized to the untreated G2019S astrocytes rather than an internal housekeeping control first.

9. In Figure 6, the authors show that treatment of G2019S BBB-chips with MEK1/2 inhibitor rescues barrier integrity defects. Treatment is also associated with lower levels of IL-6 and IL-8 cytokine secretion. Do blocking antibodies against IL-6 or IL-8 have any effect on barrier integrity in untreated G2019S BBB chips?

Minor concerns

1. It is again unusual to compute SEM or t-tests with only 2 measurements per group as in 5B
2. Figure 6D, BBB cells appear to have fully detached, Figure 3A,D vascular cells also appear to have detached or simply not formed a competent vessel.

Reviewer #3 (Remarks to the Author):

This manuscript uses iPSCs harboring a Parkinson's relevant mutation in LRRK2 to identify potential connections between astrocyte inflammation and blood-brain barrier dysfunction. The work represents a unique approach for using iPSC models to understand astrocyte-BMEC links in disease, and I think it could be a valuable addition to the literature. My main critiques are centered around data interpretation. First, I do not think the astrocyte phenotyping was done properly if the culture medium contained serum, which causes baseline astrocyte reactivity -- this would confound interpretations based on the genetic mutation. Second, some of the phenotypes observed in the vessel model are difficult to put into context. It is not clear if the vascular distortion that causes permeability increases is physiologically relevant and meaningful. This dampens enthusiasm for the significance of the work. These comments and others are elaborated below.

- 1) The experiments in Figure 1I-J and Figure 4 are interesting but need to be better contextualized. The method section of this manuscript says that after derivation, astrocytes were cultured in a “chemically defined astrocyte medium” from ScienCell. However, there is no chemically defined astrocyte medium sold by this company – rather, their medium is supplemented with 2% FBS, which has two problems. First, FBS will contain cytokines that impact the proteome array. Second, astrocytes become reactive in FBS, and this could influence their gene expression and protein secretion. It would be more appropriate to use a serum-free differentiation strategy to ascertain gene expression and protein secretion under a quiescent astrocyte phenotype in both genetic backgrounds.
- 2) Although gene expression data in the BMEC-like cells are useful for verifying endothelial identity, the authors should also immunostain for VE-cadherin and PECAM-1 for explicit confirmation.
- 3) Raw data should be included for Figure 2F-H similar to Figure 3A so that readers can properly judge the quantifications.
- 4) The data in Figure 3 should be accompanied by standard Transwell experiments. While I can appreciate the use of the microfluidic device to add three-dimensional elements and shear stress, the identification of monolayer gaps in Figure 3D hinders interpretation of barrier integrity outcomes. The authors would have no issues achieving a uniform monolayer on Transwell filters, and since the microfluidic chip is a non-contact model between the BMECs and astrocytes, a Transwell setup would complement these outcomes. An alternative would be if the authors could acquire human tissue or mouse tissue from a Parkinson’s model to show similar enlargement of vessels with low cell coverage. Otherwise, the physiological relevance of these outcomes is unclear.
- 5) Similar to my above comment, Figure 6 should include an analysis of BMEC coverage to mirror the phenotype observed in Figure 3.
- 6) Does MEK/ERK inhibition have any effect on the BMECs alone, with or without the G2019S mutation?

Reviewer #1 (Remarks to the Author):

In this work de Rus Jacquet et al examined the role of the LRRK2 Parkinson's disease (PD) associated mutation G2019S in determining proinflammatory changes that affect the integrity of the blood brain barrier (BBB). To this end, they employed patient induced pluripotent stem cells (iPSCs) and established a microfluidic 3D human BBB-chip. Using this device, the authors show that LRRK2 G2019S leads to inflammatory changes in astrocytes and defects in the formation of a functional BBB in vitro.

This work is relevant for the field as it shows that human iPSC organ-on-a-chip models can be employed to investigate BBB dysfunction in the context of human disease. However, the i) relevance and contribution of the current data to our further understanding of LRRK2-PD related mechanisms, and ii) the novelty of the chip should be better addressed. The major issues are as follows:

Major Issues, Comment #1

The novelty and the advantages of the chip over existing models is unclear and should be better addressed (PMID: 31173718; PMID: 34625559). The authors claim "The distinct advantage of this platform over other existing 3D models is the replacement of a membrane physically separating the vascular and brain compartments with an ECM scaffold that supports vessel formation, cell migration and free diffusion of secreted factors". However, in the current study the authors do not investigate cell migration within their chip model. Furthermore, the chip described by Vatine et al contained a porous membrane and allowed the investigation of diffusion of factors and their impact on iPSC-neurons that were co-cultured in the brain channel (PMID: 31173718). Indeed, the addition of neurons within the chip described by de Rus Jacquet et al would have further strengthened the relevance of the described mechanisms in LRRK2-PD neurodegeneration.

Re: The reviewer makes a good point that porous membranes used in the study by Vatine et al. allows for the diffusion of factors and the study of their impact on co-cultured cells, and we are not challenging the validity of this excellent BBB model. We only present an alternative technology that does not contain a physical barrier, and therefore offers new possible readouts. For example, new data collected for this resubmission shows that the membrane-free system allows us to quantify changes to vessel diameter in BBB^{G2019S} compared to BBB^{Ctl} (**New Figure 3M, new Figure 6K**), and these measures would not have been possible in a platform consisting of a semi-rigid scaffolding membrane. However, because the purpose of this study is not to compare existing BBB-chip technologies, we decided to remove this discussion from the main text. Regarding the inclusion of neurons in the BBB-chip, we previously showed that LRRK2 G2019S astrocytes induce neuron atrophy¹, and we chose not to include neurons in the model in order to control for confounding factors that could arise from a neuronal response to mutant astrocytes.

Comment #2

The study is also limited by the fact that key experiments have been performed using only one isogenic cell couple without further validation (i.e. Protein array in Figure 2I). Given the high variability between the two G2019S mutant lines (see for instance cytokine production in Figure 4 and differentiation yield), additional lines should have been included.

Re: We appreciate the reviewer's suggestion and we are now providing new data generated using additional iPSC lines. We repeated the experiments on which the main conclusions of the study are based, and the new replicates are shown in Figure 1K, Figure 3, Figure 6, and Supplementary Figure 3. In Figure 1K, we confirm the downregulation of angiogenesis-related proteins in a second iPSC pair. In Figure 3, we validate the barrier integrity phenotype of BBB^{G2019S} vs. BBB^{Ctl} using a third iPSC pair. In Figure 6, we confirm that ERK-inhibition decreases IL-6 and IL-8 secretion, and rescues barrier integrity in BBB^{G2019S} vs. BBB^{Ctl} using a second iPSC pair. The information

relative to the iPSC lines included in this study (e.g. provenance, sex etc.) is provided in our new Supplementary Table 1. Overall, the addition of new replicates and a new iPSC line confirms the data provided in the initial submission and strengthens our conclusions.

In addition, we confirm that astrocyte differentiation yields are very consistent across all lines, as shown in the new Supplementary Figure 2D. The reviewer pointed out experimental variability in Figure 4A-B. In this figure, we quantified gene expression levels of inflammatory markers and we observed differences between LRRK2 G2019S and control astrocytes. We agree that the isogenic pair appears to show greater changes in gene expression levels compared to the non-isogenic pair, but this difference is attenuated in Figure 4C. Overall, the trends remain consistent across both lines, with cytokines being consistently overexpressed and oversecreted by LRRK2 G2019S astrocytes.

Comment #3

As described below, the authors performed a meta-analysis that shows the dysregulation of specific genes and pathways in LRRK2 mutant astrocytes. However, none of these candidates is further investigated in subsequent functional analysis. Similarly, the protein array performed in isogenic LRRK2 iPSC-astrocytes show very minor differences. The graph in 1J shows significant differences only for 3 candidate proteins. Based on the data, how can the authors conclude that the G2019S mutation is linked to angiogenesis-related changes?

Re: Regarding the protein array in Figure 1, we increased the number of biological replicates and confirmed the results using a second iPSC pair. We now provide an updated Figure 1I-K with 9 significantly dysregulated angiogenesis-related factors.

Among the meta-analysis data provided in Figure 1, changes to inflammation-related factors caught our attention because previous studies suggest that inflammatory factors can alter angiogenesis and BBB function²⁻⁵. Given that (i) IL-8 and IL-6 are inflammatory cytokines (IL-6 is the most upregulated genes in the “inflammation” GO category, see **Figure 1E**), (ii) inflammation has been suggested to affect the vasculature and (iii) patients with PD display extensive neuroinflammation and BBB dysfunction, we wanted to better understand the relationships between astrocytes, cytokine secretion and BBB function. Therefore, we extensively studied IL-8 and IL-6 secretion profiles in LRRK2 G2019S astrocytes (**Figures 4 and 5**) and assessed their effect on barrier integrity (**new Figure 4H, I**). In particular, this new data added to Figure 4 shows that (i) supplementation of IL-6 and IL-8 in the brain compartment triggers loss of barrier integrity in BBB^{ctrl} vessels and (ii) treatment of BBB^{G2019S} with an IL-8 neutralizing antibody partially rescues paracellular permeability. This partial restoration of vessel function suggests that multiple factors act synergistically to trigger the observed phenotype. To support this hypothesis, inhibition of the ERK cascade – which regulates the expression of several signaling molecules, including cytokines, fully restored BBB integrity.

Nonetheless, we had previously tested angiogenesis-related candidates for which the data was not included in the first version of the manuscript. We now provide the results in a new Supplementary Figure 4:

Apolipoproteins: Our meta-analysis showed dysregulation of *APOLD1* (**Figure 1G**) and *APOL3* (**Figure 1E**), two genes coding for apolipoproteins. This data therefore suggests that astrocytes with the LRRK2 G2019S mutation present with changes related to this broad family of signaling molecules. Apolipoproteins form a large group, and apolipoprotein E (ApoE) is one of the most studied members due to its implications in Alzheimer’s disease, where loss of ApoE results in BBB leakage⁶. We therefore evaluated the secretion of ApoE by our isogenic astrocyte pair and found a significant loss of ApoE release in the conditioned media, in both monolayers and BBB-chips (**New Supplementary Figure 4A, B**). We therefore assessed whether supplementing the brain compartment with ApoE could restore barrier integrity and found a significant reduction of passive paracellular permeability at day 4. However, this effect was not maintained until day 6, as P_{aap} values of ApoE-treated BBB^{G2019S} increased from 4.7×10^{-6} cm/s (Day 4) to 6.7×10^{-6} cm/s (Day 6), thus reaching similar P_{aap} values as untreated BBB^{G2019S} (**New**

Supplementary Figure 4C-F). These observations suggest that other mechanisms are driving the phenotype. Furthermore, ApoE supplementation did not improve rhodamine retention in the vessel.

Wnt signaling: Brain vascularization is a complex process regulated by pro-angiogenic Wnt proteins⁷ and we initially hypothesized that a reduced BBB^{G2019S} function could be mediated by a lack of pro-angiogenic factors. We therefore attempted to rescue vessel formation and barrier integrity by activating Wnt signaling in the BBB^{G2019S} vessels. To do so, we treated the vessel compartment of the BBB^{G2019S} model with CHIR99021, a small molecule activator of Wnt signaling and measured vessel permeability to 4.4 kDa dextran-TMRE and rhodamine dyes. The data show that CHIR99021 improves passive paracellular permeability at day 6, and p-glycoprotein efflux is improved at day 4, but not at day 6 (**New Supplementary Figure 4G-J**). As a result, loss of angiogenic signals does not appear to be the primary mediator of BBB^{G2019S} dysfunction.

Comment #4

The comparison of the PD findings to AD APOE and AD PSENΔE9 results is interesting. However, as the dysfunction of BBB has emerged as a potential pathogenetic factor in AD, these results are quite surprising. To conclude that their findings are rather specific to PD, it would be valuable to consider adding additional iPSC findings with genetic alterations associated with AD. This will be particularly important in consideration of the high variability among different lines and differentiation protocols (overall their findings seem to be largely driven by the study by Di Domenico et al.). This issue should be further discussed; are different mechanisms (genotype specific) involved in BBB dysfunction in PD vs AD? Is there evidence of an increased BBB disruption in LRRK2-PD patients?

Re: The reviewer makes an interesting point, however the study is not focused on PD vs. AD differences. Further documenting changes to AD-derived astrocytes would take away from the main message of this work, which is the impact of PD astrocytes in BBB function. As a result, we removed this data from the manuscript.

Comment #5

Figure 6D: regarding the image with G2019S astrocytes, one may wonder whether this is very representative of their findings. The highlighted white area shows the failure of vessel formation; however, previous images do not indicate such a significant change. Also, Western blot results (Figure 3E) show no differences in any of the BBB markers that could explain such a change. With this regard, the authors link the BBB dysfunction to the ERK pathway and performed rescue experiments measuring the VE-cadherin, claudin 5, ZO-1 levels. However, in Figure 3E, none of these shows significant changes in LRRK2-G2019S astrocytes.

Re: Images in Figure 6D (now **Figure 6E**) have been replaced to better represent changes to vessel morphology in the BBB^{G2019S}. In addition, we thank the reviewer for this question regarding the correlation between tight junction marker levels and barrier function. We hypothesized that rescue of barrier integrity by ERK inhibition could be associated with a higher expression level of tight junction proteins. To address this hypothesis, BBB^{G2019S} were treated with the small molecule ERK inhibitor (PD0325901) and VE-cadherin, claudin 5, and ZO-1 protein levels were measured by western blot. We found that the protein levels for these markers were not significantly increased compared to untreated BBB^{G2019S}, suggesting that rescue of barrier integrity is likely mediated by other factors. This data points towards a model where BBB dysfunction could occur independently of tight junction protein levels. In the context of BBB^{ctl} vs. BBB^{G2019S}, it appears that BMEC-like cells efficiently produce tight junction proteins but they may fail to self-reorganize into a fully formed vessel, suggesting that vessel formation is mediated by complex factors involving other aspects than tight junction protein expression. To the best of our knowledge, current studies utilizing microfluidic models have not attempted to document how tight junction protein expression varies depending on environmental cues. We believe this observation is valuable to the field and contributes to building a better understanding of the molecular dynamics governing vessel integrity.

Additional comments, Comment #6

Figure 1 (Meta-analysis of RNA sequencing data sets): the authors should validate the DEG emerged from the metanalysis and relevant to the angiogenesis/immune dysregulation pathways in the isogenic LRRK2 astrocytes employed in the current study.

Re: We apologize if the results and methods were not explicit enough, but the isogenic iPSCs and iPSC-derived astrocytes included in the current study are exactly the same as those used to produce one of the RNA-sequencing datasets included in the meta-analysis (de Rus Jacquet et al. 2021, *eLife*). The first author in these two studies is the same and produced the astrocytes following identical protocols⁸ and the same isogenic line. To address the reviewer's comment, we refer to Figure 1J (validation of RNA-sequencing) and Figure 4 (validation of the immune dysregulation). These figures confirm the dysregulation of the angiogenesis and immune pathways at the gene expression and protein secretion levels in the isogenic and non-isogenic lines used in our study.

Comment #7

Figure 1B: the authors should show the list of DE expressed genes used for their analyses. Based on the graph and the text, they performed GO analyses based on 21 genes and split them based on up-and down-regulated. It is unclear how many genes went into the analyses? Also, the number seems quite low for GO analysis.

Re: We thank the reviewer for this opportunity to clarify our methodology. The dataset used to generate the gene ontology analysis was produced as follows: (1) DEG were listed for each LRRK2 G2019S vs. control RNA-seq study included in this article, (2) all non-identical DEG were combined into a single file and (3) the combined DEG list was imported into DAVID to produce the gene ontology assessment. We now provide a **new supplementary file** listing all 4,008 DEG used for the gene ontology analysis (2,007 genes downregulated, 2001 genes upregulated).

Comment #8

Figure 1I, J: we understand that the protein array experiment has been performed on one isogenic iPSC couple. To strengthen the validity of these findings, results from additional lines should be provided. Validation with Western Blot/Elisa should be provided.

Re: According to the reviewer's suggestion, we now provide additional data showing the angiogenesis protein array performed using an additional iPSC pair (**New Figure 1K**). The new data confirm and reinforce the findings that the LRRK2 G2019S mutation reduces the expression/secretion of angiogenesis-related molecules by astrocytes.

Comment #9

Supplementary Figure 1: the authors should improve the characterization of the iPSC-derived astrocytes including differentiation yield, inflammatory profile (relevant to the current study), and representative images for all cell lines.

Re: The differentiation yields and representative images for all cell lines are now available in the **new Supplementary Figure 2**. The inflammatory profile is available in Figures 4 and 6.

Comment #10

Figure 2C: the authors claim a 22-fold increase in VE-cadherin in BMEC-like cells; however, to have a better comparison, they should provide the Western blot for hBMEC (as they did for the gene expression in Figure 2B). The authors state “All samples were run on the same gel, and the bands displayed under the graphs are from the same membrane”; however, they only show individual cropped bands. Also, the variability seems to be extremely high and makes wonder about the robustness of the protocol.

Re: We thank the reviewer for this suggestions and we have now collected western blot data comparing the expression levels of BMEC markers in BMEC-like cells vs. human primary BMECs. The bands displayed in **the new Figure 2C** are from the same membrane. We selected the following four markers: VE-cadherin, claudin 5, ZO-1 and CD31, and the data is shown in the new Figure 2C. These western blots were performed using monolayers of BMEC-like cells grown on Geltrex and not on a BMEC-selective matrix such as collagen/fibrinogen. Therefore, the variability observed in a subset of markers could be related to the presence of a mixed population of cells with varying levels of BMEC-specific markers. However, we refer the reviewer to Figure 3F-I where BMEC-like cells form a 3D vessel on a BMEC-selective matrix, and variability is minimal between independent replicates.

Comment #11

Figure 2C: overall, it is difficult to assess the efficiency of differentiation protocol and the validity of the chip model. The stainings for VE-cadherin and Occludin display an unexpected nuclear pattern. Furthermore, the authors should further characterize the vascular channel with additional markers (e.g. claudin 1, CD31). Markers as the BBB glucose transporter GLUT-1 should also be assessed. Similarly, the differentiation of iPSCs into brain microvascular endothelial-like cells should be further characterized.

Re: Additional validation of the BMEC-like cell differentiation is now available in the **new Figure 2C and 2E**, where we illustrate expression of VE-cadherin, claudin 5, ZO-1, GLUT-1 and CD31. We show that the BMEC-like cells express levels of VE-cadherin, claudin 5, and ZO-1 that are comparable to those of primary human BMECs. Our data suggests that the iPSC-derived BMECs do not express CD31, in accordance with previous publications discussing the mixed endothelial/epithelial phenotype of these cells⁹. To reflect this observation and avoid misinterpretation, we use the term “BMEC-like cells” and not “BMECs” throughout the manuscript. The reviewer questions a possible nuclear pattern of VE-cadherin and occludin staining in Figure 2D. The images used to produce Figure 2D are displayed below as single-staining (without DAPI nuclear stain), and we only see minor instances of nuclear staining in occluding (white star), and no nuclear staining in the VE-cadherin image. For clarity, images in Figure 2D showing VE-cadherin/DAPI and Occludin/DAPI staining have been replaced with images lacking the DAPI counterstain.

Occludin

VE-cadherin

Comment #12

Figure 4: the authors investigate the reactive phenotype of G2019S astrocytes. Did this phenotype also emerge from their metanalysis?

Re: The meta analysis provided in Figure 1 shows inflammation-related components as a main dysregulated GO term (**Figure 1C**). Examples of GO terms associated with upregulated genes include “Response to lipopolysaccharide”, “Negative regulation of JAK-STAT cascade”, “Cytokine-mediated signaling pathway”. As detailed in our response to comment #6, the isogenic astrocytes used in the current study are the same as the astrocytes published in the article by *de Rus Jacquet et al.*¹ and included in the meta-analysis. The current manuscript therefore validates the inflammatory phenotype of LRRK2 G2019S astrocytes using both RNA-seq (**Figure 1**) and biochemical approaches (**Figure 4**).

Comment #13

The authors identified a significant difference in IL-6 and IL-8 secreted by G2019S astrocytes. Besides their role in neurovascular function, is there any specific known association between these cytokines and G2019S-PD?

Re: The reviewer asks an important question, and the text below has been added to our discussion:

Notably, the secretion of IL-6 and IL-8 by immune cells has been reported in studies focusing on the inflammatory response in people with PD. A clinical study found that increased IL-8, MCP-1 and MIP-1- β levels in the serum of people with LRRK2-PD correlated with severe motor and non-motor clinical subtypes¹⁰. In a study by Cook et al, patient-isolated circulating immune cells secreted increased levels of IL-6 upon immune challenge, and LRRK2 expression was increased in B cells, T cells and CD16+ monocytes¹¹. In another report, iPSC-derived monocytes and macrophages harboring the LRRK2 G2019S mutation were stimulated with toll-like receptor agonists, and the investigators observed increased cytokine secretion (e.g. IL-6, IL-8) in the mutant cells compared to isogenic controls¹². The increased release of IL-6 and IL-8, associated with changes to LRRK2 expression or kinase activity, corroborates our findings related to LRRK2 G2019S astrocytes. An interesting report suggests that LRRK2 does not regulate cytokine release in the same manner in all cell types¹³. In their work, the authors show that iPSC-derived microglia carrying the LRRK2 G2019S mutation decrease their secretion of IL-6 and IL-8 upon LPS stimulation¹³. While the specific biological consequences of cytokine secretion on PD-related pathological features remains to be further investigated, the present study proposes a link between inflammation and alterations at the BBB using a human model recapitulating the architectural complexity of the neurovascular unit.

Comment #14

Suddenly, the authors jump to the analysis of the MEK/ERK pathway, which however did not emerge as one of the candidates in their metanalysis nor in the protein array investigation. It would have been more congruent to focus on one of the newly identified candidates.

Re: We thank the reviewer for this question and opportunity to clarify our investigation of the MEK/ERK pathway, which indirectly emerged as one of our candidates. As detailed in our response to comment #3, we initially studied promising candidates based on their pro-angiogenic properties. However, these showed moderate effects, suggesting that alteration of other biological processes may better explain the phenotype induced by LRRK2 G2019S astrocytes. We therefore identified a kinase cascade, the MEK/ERK pathway, that can simultaneously modulate a large number of factors associated with both BBB dysfunction and the inflammatory response. To further address the concern from this reviewer, we provide a new computational analysis in **Figure 5A**. This analysis unveils that transcription factors downstream of the ERK cascade target differentially expressed genes belonging to the GO categories angiogenesis, inflammation and cell adhesion. Notably, the top 2 ERK-related

transcription factors, identified in Figure 5A, target 55 to 85 % of the DEG across the three GO categories. Therefore, this data positions the MEK/ERK pathway as a strong candidate implicated in BBB^{G2019S} dysfunction.

Overall, the current study addresses a significant gap in our understanding of the contributions of astrocytes to BBB function in PD, and the identification of the MEK/ERK pathway as a seemingly key signal that regulates two essential astrocytic roles, BBB function and inflammation. In addition, the MEK/ERK pathway was found to impact the viability of LRRK2 G2019S iPSC-derived neurons¹⁴ and, together with the current study, these findings strengthen the implication of MEK/ERK regulatory functions in PD-related processes.

Comment #15

In the discussion, the authors refer to inflammation, infiltration of peripheral immune cells and impaired BBB maintenance in human tissue but they don't really discuss findings from the primary literature. The study would acquire much more impact if complemented with validation results from post-mortem tissue of LRRK2 carriers. At least, it would be important to have a more detailed discussion about their iPSC findings compared to results from patient tissues samples.

Re: We appreciate this suggestion and have now substantially improved the manuscript with a new figure (**new Figure 8**) confirming our findings of vascular impairments in the human *substantia nigra*, using post-mortem brain tissue from patients with PD. A previous studies suggested the presence of morphological changes to the brain vasculature in postmortem PD tissue¹⁵, and we verified this data using our own independent cohort. The new data demonstrate a lower coverage and decreased number of blood vessels as well as an enlargement of vessel diameter in the *substantia nigra* of people with PD. This observation corroborates our data showing that LRRK2 G2019S astrocytes increase vessel width in the BBB-chip (**new Figure 3M**).

Comment #16

Given the ongoing clinical trials with LRRK2 kinase inhibitors, it would be valuable to show whether the mechanisms described here are LRRK2 kinase dependent. This would also strengthen the validity of the model for drug testing.

Re: The reviewer makes an excellent point and we therefore collected additional barrier integrity data after BBB^{G2019S} treatment with the LRRK2-IN-1 small molecule kinase inhibitor. For these experiments, we produced BBB^{G2019S} chips and treated the brain compartment with 5 μ M LRRK2-IN-1 (**new Figure 6L, M**). The results show that pharmacological inhibition of LRRK2 kinase activity did not improve barrier integrity. This finding is interesting because our model leveraged iPSCs harboring a LRRK2 kinase gain-of-function mutation, thus suggesting that the phenotypes are LRRK2 kinase-dependent. This unexpected finding could be explained by the limitation of a pharmacological approach. Treatment of the brain compartment did not specifically target the mutant astrocytes, and LRRK2 inhibition also affected the pericytes and the abluminal part of the vessel wall. Given the importance of LRRK2 in cell-cell communication¹, a broad, non-targeted LRRK2 inhibition appears inefficient to restore barrier formation. Alternatively, LRRK2 kinase activity is likely to impact the early stages of astrocyte differentiation, which could mediate the BBB impairment observed in the chip regardless of kinase inhibition at the time of the experiment. To address this complex issue, future studies could take advantage of a LRRK2 kinase dead variant, and compare vessel formation and barrier integrity in this genetic model.

Comment #17

The authors should show levels of LRRK2 for all the iPSC lines astrocytes.

Re: The protein levels of LRRK2 for the three lines of astrocytes used in this study are now provided in the **new Supplementary Figure 2C**.

Minor issues, Comment #18

Figure 2B: It looks like there is a typo as the heatmap labelling shows CHD5 and not CDH5.

Re: We thank the reviewer for noticing this typo.

Reviewer #2 (Remarks to the Author):

General comment #1. *In this work Rus Jacquet et al seek to interrogate the role of PD astrocyte induced neuroinflammation on the integrity of the BBB using an iPSC derived model and microfluidic device. Whether BBB integrity is impaired in PD remains controversial (Fujita, K. et al. Blood-brain barrier permeability in Parkinson's disease patients with and without dyskinesia. J Neurol 268, 2246-2255) and early in disease there is not as much evidence of neuroinflammation as seen in other neurodegenerative diseases such as ALS and AD although there is inflammation late in disease. Thus, the rationale for their study is not quite as strong as the authors suggest and they need to cite a wider range of papers both for and against the ideas they present (or as challenges for the study).*

Re: To address the reviewer's concern, we now discuss a broader range of publications reported BBB impairments in people with PD. Notably, we discuss the study by Fujita et al. in the context of the other publications in the field, and we note that the authors themselves attempt to explain why their findings are in contradiction with other studies reporting BBB impairments and which include the following explanation:

"We speculate that for very low K1 values the ⁸²Rb bolus technique may not be able to estimate very small changes in influx accurately because of the short half-life" (...) "Given the limited number of subjects in the current study and the complex nature of the imaging protocol, the results may not be generalizable to the PD population at large."

In addition, we further investigated vascular changes in the postmortem *substantia nigra* of people with PD, and we report decreased vascular coverage along with increased vessel diameter compared to control tissue (**New Figure 8**). This data complement previously published clinical studies and strengthen a vascular hypothesis to PD pathogenesis.

The new text reads as follows:

Introduction:

Evidence of capillary leakage in PD has been demonstrated in brain imaging studies ^{16,17}, abnormal perivascular deposit of serum proteins was shown using histological analyses of postmortem brain tissue of PD patients ^{18,19}, and albumin/IgG levels were increased in the cerebrospinal fluid of patients compared to controls ²⁰. In parallel, increased angiogenesis was observed, a process that could lead to the formation of immature blood vessel with weaker blood-brain protective properties ^{21,22}. The cause of BBB permeability in PD has not been elucidated, but evidence suggests that pro-inflammatory mediators, including cytokines, could affect its integrity ^{2,3}.

Discussion:

BBB dysfunction has been documented in patients with neurodegenerative diseases, including PD ²³. More specifically, clinical studies using medical imaging technologies such as Magnetic Resonance Imaging and Positron Emission Tomography have documented capillary leakage in the brains of patients with PD ^{16,17}. For example, dynamic contrast enhanced magnetic resonance imaging (DCE-MRI) is a technology used to measure subtle changes to BBB integrity, and it was used to identify increased leakage of gadolinium in the basal ganglia of people with PD ¹⁶. A contradicting study, however, reported no difference in the striatal permeability to the potassium analogue rubidium-82 (⁸²Rb), which was used to monitor disruption of tight junctions ²⁴. The authors suggest that their methodological approach might not detect mild changes in ⁸²Rb influx. In another study, investigators

utilized positron emission tomography to assess brain infiltration of [11C]-verapamil, a substrate of the efflux transporter p-glycoprotein, and showed increased uptake in the midbrain of people with PD¹⁷. In addition to imaging studies of barrier integrity, histological analyses of postmortem human brain samples of patients revealed compromised striatal BBB, characterized by abnormal deposits of serum proteins in the brain parenchyma and erythrocyte extravasation¹⁸. Collectively, these studies support the theory that barrier integrity is reduced in the brains of people with PD. These BBB changes may also be accompanied by angiogenesis-related biochemical changes occurring in the brain of patients, such as increased VEGF²⁵, which could be involved in vascular remodeling. These collective findings that potentially toxic circulating factors could breach the BBB and enter the brain parenchyma to affect neuronal survival support the hypothesis of a peripheral contribution to PD etiology. To further corroborate this idea, patients undergoing deep brain stimulation of the subthalamic nucleus (STN-DBS) tend to exhibit an attenuation of motor features and slower disease progression, and these important changes correlate with improved brain microvasculature. For example, STN-DBS upregulated ZO-1, claudin 5, VE-cadherin, and occludin levels, as well as decreased microglia density compared to non-STN-DBS patients¹⁹.

General comment #2. *In general, this work is hindered by the relatively few iPSC lines used in these studies and a poor characterization of the iPSC derived BBB endothelium in addition to non appropriate statistical tests (t-test when should be ANOVA).*

Re: To address the reviewer's concern regarding the number of iPSC lines used in this study, we validated our key results using an additional LRRK2 G2019S patient derived iPSC line. We refer to our response to a similar concern by reviewer 1, comment 2. In addition, all statistical analyses have been verified in collaboration the statistics platform at the CHUL de Quebec-Université Laval. In his concern regarding the use of t-test vs ANOVA, we suspect the reviewer refers to Figure 1J,K Figure 4A,B and Figure 5D-E. In these Figures, we combine the protein (Figure 1J,K) or gene expression (Figure 4A,B and 5D-E) levels for several markers into a single graph. The value for each protein or gene of interest is displayed as the normalization (i.e ratio) of LRRK2 G2019S vs control astrocytes. We then performed a one sample t-test because we want to assess statistically significant changes for each individual protein/gene, between control and LRRK2 G2019S conditions. Our statistician collaborator confirmed that an ANOVA would not be appropriate here, because we do not aim to assess how the fold change for each protein/gene compare to one another.

General comment #3. *Transwell studies to show increased permeability with the LRRK2 astrocytes and TEER measurements of the iPSC derived BBB endothelial cells would help to address some of the basic characterization concerns about the iPSC derived BBB and interplay with PD astrocytes.*

Re: As suggested by the reviewer, we performed transwell studies to determine whether LRRK2 G2019S astrocytes disrupt the formation of an endothelial monolayer (**new Figure 3E**). In these experiments, the membrane of the transwell (vascular-like compartment) was plated with a monolayer of control BMEC-like cells, and the bottom chamber (brain-like compartment) was plated with control pericytes and either control or G2019S astrocytes. The growth medium in the top chamber was supplemented with 4.4 kDa dextran-TMRE, and medium in both compartments was collected after an overnight incubation. The fluorescence intensity was then measured and plotted as the ratio of G2019S vs. control BMEC-like cell permeability. The data show that BMEC-like monolayers exposed to mutant astrocytes, similarly to the BBB^{G2019S} chip, are more permeable to 4.4 kDa dextran. This observation confirms that LRRK2 G2019S astrocytes alter the function of BMEC-like cells.

General comment #4. *Other major concerns include the integrity of the vascular cells in the microfluidic devices as well as the important positive control of direct inflammation challenge of the control chip barrier properties with LPS (or similar inflammatory cytokine) to establish if the impaired barrier properties are indeed directly related to*

inflammation and not some other mechanism. Additionally, several studies lack sufficient data for convincing statistical analysis, and the overall clarity of how many independent differentiations, experiments is lacking.

Re: We agree that an inflammatory challenge is an important control to validate that the vessels are responsive to this trigger. We would like to refer the reviewer to our **Figure 4H**, in which we already provide data showing that inflammatory challenges by IL-6 and IL-8 reduce BBB^{CTL} integrity. We now complement this data by showing the partial rescue of barrier function in BBB^{G2019S} treated with an IL-8 neutralizing antibody (**new Figure 4I**). Taken together, these findings support a role for inflammatory factors in BBB dysfunction and corroborate a recent study published in *Nature Communications*⁵ showing that reactive astrocytes weaken the BBB. However, non-inflammatory factors may also act synergistically to weaken the endothelial vessel, and we address this possibility in the main text of the manuscript. Furthermore, we increased the number of biological replicates to strengthen the power of our statistical analyses (**new Figures 1J-K, Figure 2, Figure 3 and Figure 6**), and we included a third iPSC line to validate key findings (**new Figure 3**). In addition, we confirm that the data provided in the original and revised manuscript were collected using BMEC-like cells originating from independent differentiations. We refer the reviewer to our statement provided in the Material and Methods section of the initial manuscript: “BMEC-like cells were newly differentiated before each BBB-chip experiment”. To clarify the number of independent astrocyte differentiations, we added the following statement to the revised Material and Methods section: “Overall, up to four independent astrocyte differentiations were performed for each iPSC lines, and each batch produced a large number of cryopreserved aliquots for long term storage and experimental use.”

Major Concerns, Comment #1

Given that the initial part of this study focuses on comparing previously published data sets, this work seems to be missing the important comparison of if all 3 differentiations made a similar product especially with the wildly different altered numbers of differentially expressed genes. Even though this is previously published data, information on the patient and control lines needs to be provided. Number of individual lines/clones in each study, Age of onset, age of iPSC generation, source tissue ect. Are the controls isogenic? Also need this data for figure 1I. It is clear now that there will always be DEGS when comparing diverse data sets like these – but validation against other data sets is the key aspect. As extracellular matrix tends to be the most variable in these studies – these genes also tend to be the most different between groups but on validation this often falls apart.

Re: The reviewer makes an excellent point, and we have confirmed that the astrocytes produced in the three papers (i.e. de Rus Jacquet (*eLife*), di Domenico (*Stem Cell Reports*) and Booth (*Neurobiology of disease*)) are comparable. We performed a bioinformatics analysis to measure the expression levels of key astrocyte markers in all three differentiations, and we found that they are comparable to one another, to human primary cortical astrocytes, and they differ from iPSCs (**new Supplementary Figure 1**). The data also show that none of the astrocytes express neuronal markers. Furthermore, we have compiled all relevant information regarding the origin of the iPSC lines used in the study, and this information is included in the revised version of the manuscript (**new Supplementary Table 1**).

Comment #2

The number of iPSC lines and clones is again a concern. It would also be clearer if the LRRK2 astrocytes were compared to an isogenic corrected control for each patient with data from at least 3 patients presented throughout these studies.

Re: To address the reviewer’s concern regarding the number of lines, we added a third LRRK2 G2019S iPSC line and validated key experiments. We refer to our response to reviewer 1, comment 2. Unfortunately, despite our best efforts, we were not able to obtain an additional isogenic pair. Therefore, the data provided in this manuscript was collected using 1 isogenic and 2 non-isogenic pairs.

Comment #3

Data presented in Figures 2B, 2G, 3A, 3C, 4E, 5B, 5F, and S2A contain only two biological replicates and are likely not sufficiently powered to perform tests of significance in all cases. Additionally, much of the data presented does not appear to be representative of independent experiments which makes it difficult to assess the reproducibility of the model or of the results generated.

Re: We thank the reviewer for bringing this point to our attention. Our experimental results have been updated to increase the number of biological replicates, and the data presented in the new figures represent a minimum of 3 independent biological replicates. We repeated the experiments on which the main conclusions of the study are based, and the new replicates are shown in Figure 1K, Figure 2, Figure 3, Figure 5C, Figure 6, and Supplementary Figure 3. In Figure 1K, we confirm the downregulation of angiogenesis-related proteins in a second iPSC pair. In Figure 2, we included additional replicates to confirm the gene expression data and the barrier integrity assays. In Figure 3, we validate the barrier integrity phenotype of BBB^{G2019S} vs. BBB^{Ctl} using a third iPSC pair and added new biological replicates to reinforce the dextran and rhodamine permeability assays. In Figure 5C we added more replicates to confirm the activation of the ERK pathway in LRRK2 G2109S astrocytes. In Figure 6, we confirm that ERK-inhibition decreases IL-6 and IL-8 secretion, and rescues barrier integrity in BBB^{G2019S} vs. BBB^{Ctl} using a second iPSC pair. In Supplementary Figure 3, we added new replicates to validate the gene expression data. We removed the initial Figure 4E because the array was too exploratory and did not further serve the study, compared to the more important gene expression (Figure 4A-C) and cytokine secretion (Figure 4D-G) data.

Furthermore, we confirm that data provided in the initial and revised manuscript were collected from independent experiments, and we made sure that the figure legends systematically report the number of independent biological replicates. We also address this concern in our response to “reviewer 2, General comment #4”.

Comment #4

The barrier integrity assays presented in Figure 2F-G show high variability between biological replicates and an inability to distinguish any difference in permeability between Fluorescein, 4.4kDa-Dex, and Rhodamine. It is unclear if the high variability is due to the method of assessing permeability (Image J quantification of fluoresce intensity changes in EVOS images) or due to high variability inherent in the model. Additional permeability assessments should be made with various sized dextrans and/or with compounds with known BBB permeabilities to determine the reproducibility and robustness of the barrier and method of assessment.

Re: To address this concern, and as suggested by the reviewer, we now provide an ELISA-based quantification showing that the BBB-chip model prevents the crossing of IgG from the vessel to the brain compartment (**New Figure 2G**).

A bigger question is - is it that an effective barrier fails to form or that the formation is delayed in the absence of the astrocytes? Only 1 timepoint at 6 DIV was investigated. Since the BMECs were newly generated how was each set of cells characterized prior to seeding in the chips? This is especially important given the variation in tight junction proteins in 2C. It is also unclear how many independent differentiations are represented in any of the figures – you don't specifically answer this.

Re: We have consistently observed that if P_{app} values reach $\sim 1 \times 10^{-5}$ cm/s at day 6, they do not improve with time. Regarding the characterization of BMEC-like cells, the reviewer points out that data presented in Figure 2C show variability in tight junction levels across different differentiations. In this experiment, we plated BMEC-like cells as monolayers on Geltrex-coated plates, and this matrix is not selective for endothelial cell-specific attachment. It is therefore possible that some cells with lower BMEC-like characteristics may adhere to the plate and increase

variability in the western blot quantifications. However, in the BBB-chip, the vessel lane is coated with fibronectin and collagen IV, and this BMEC-selection step allows us to reduce the variability in tight junction level quantifications, as shown in Figure 3F-I. In light of the above findings and the extensive literature showing the central role of astrocytes in BBB formation²⁶⁻²⁹, we therefore propose that the absence of astrocytes prevents formation of a tight BBB in our model.

Comment #5

In Figure 3 B,C the use of a 2 way ANOVA is misleading. Since each fluorescent dye is examining a separate aspect of the barrier property, these should be treated as separate studies. The appropriate comparison for each dye would be a one-way ANOVA with a Tukey or Sidak multiple comparison test. It is also confusing as to why several conditions only have 2 biological replicates shown (ie 2B control #1 vs G2019S #1) and how this statistical comparison was performed.

Re: The reviewer refers to our study of 4.4 kDa dextran-TMRE and rhodamine permeability in BBB^{G2019S} vs. BBB^{CTL} chips. We would like to point out that each dye was already analyzed separately and plotted in two different graphs (4.4 kDa dextran-TMRE in Figure 3B, and rhodamine in Figure 3C). Each graph shows the P_{app} values recorded for BBB^{CTL} vs BBB^{G2019S} prepared using independent iPSC lines, and we confirmed with our Institute's statistical platform that a 2-way ANOVA is an appropriate analysis for these graphs. Furthermore, we collected additional biological replicates to these experiments, and we validated the findings using a third and independent iPSC pair.

Furthermore, the authors observe increased permeability in BBB-chips co-cultured with LRRK2 G2019S astrocytes, but no change in tight junction proteins. However, they do note increases in BMEC-like cell size in BBB-G2019S chips (Figure 3I-K). If similar numbers of BMEC-like cells are in Control and G2019S chips, the larger cells likely result in larger vessels with more surface area. Judging by the shape and size of vessels shown from above in Figure 3A and in cross section in Figure 3D, there does seem to be some differences in vessel area between Control and G2019S. Have the authors accounted for differences in vessel surface area in their permeability measurements (or shown that there is no difference in vessel surface area) due to the increased cell size seen in BMEC-like cells in G2019S chips? If not, differences in area can lead to differences in permeability that could be misinterpreted as differences in barrier.

Re: The new manuscript now provides a quantification of vessel width, and we found that BBB^{2019S} vessels are larger than BBB^{CTL} vessels (**New Figure 3M**). Importantly, we validated these findings using post-mortem sections of human *substantia nigra* and confirmed that PD patients (n=5) display vessel enlargement compared to controls (**new Figure 8**). The reviewer asks an interesting question, suggesting that increased vessel surface area could translate into increased diffusion of the dyes. Despite our best efforts, we have not found published literature demonstrating a relationship between BMEC size and barrier property. However, we understand the reviewer's question and we would happily include additional information if the reviewer wishes to share the DOIs of relevant studies.

Comment #6

The authors conclude on Line 339 that "formation of leaktight vessels requires the presence of both pericytes and astrocytes (Figure 2F-H)". However, the data only supports the requirement of astrocytes in the formation of leaktight vessels. No "Astrocyte Only" condition has been included in Figure 2G-H to determine the necessity of pericytes in the model. It is unclear if the pericytes are required for vessel formation or if they are involved in any of the crosstalk between endothelial cells and astrocytes in the model.

Re: To answer the reviewer’s question, we performed a new experiment that only included astrocytes in the brain channel. Under these conditions, the newly formed vessels are equally leaktight compared to vessels formed in the presence of both astrocytes and pericytes. The new data is included in the manuscript (**new Figure 2J, K**), and the main text has been updated to reflect that astrocytes alone are sufficient to promote functional vessel formation.

Comment #7

The gene expression changes presented in Figure 4A and 4B, which are used to support the claim that G2019S mutation is associated with upregulation of A1, A2, and pan-reactive astrocyte genes, show almost no overlap and often conflicting direction of gene expression change in the G2019S vs. Control comparison across the two sets of lines tested. A third set of G2019S vs healthy control cells should be included, or replication of gene expression changes from independent experiments should be included to determine if the expression changes are consistent or just highly variable from differentiation to differentiation.

Re: We thank the reviewer for this opportunity to clarify the data. The original dataset provided in Figure 4A and 4B show the combined gene expression levels replicated across independently differentiated astrocytes (**Figure 4A**), or astrocytes differentiated from three different clones of the same patient line (**Figure 4B**). While these two iPSC lines may show some differences in the fold change levels of gene expression for a subset of markers, overall they display a similar trend suggestive of increased expression of inflammation-related genes by LRRK2 G2019S astrocytes.

Comment #8

Given that the ERK pathway is central to proliferation, it is unclear how the total number of cells is controlled for in the figure 5 ERK inhibitor studies? It seems with both the gene expression and ELISA data are only normalized to the untreated G2019S astrocytes rather than an internal housekeeping control first.

Re: To clarify our data analysis, we would like to point out that the gene expression data is normalized to the housekeeping control actin, before being normalized to untreated G2019S. We refer the reviewer to the Material & Methods section:

“Relative mRNA levels were calculated for each gene using the formula:

$$2^{-\Delta\Delta Ct} = 2^{-((Ct, TG - Ct, \beta actin)_{LRRK2 G2019S} - (Ct, TG - Ct, \beta actin)_{WT})}$$

where “Ct, TG” represents the cycle threshold (Ct) for the target gene (TG), and “Ct, β actin” represents the cycle threshold for the loading reference *ACTB* (β -actin)”.

Regarding the ELISA, the data is not normalized to an internal control. However, the cytokine secretion data (ELISA, Figure 5H-K) depict the same trend as the gene expression data (qPCR – normalized to housekeeping control, Figure 5F). Furthermore, Figure 5K demonstrates that ERK inhibition does not significantly change IL-6 secretion in the isogenic pair.

Comment #9

In Figure 6, the authors show that treatment of G2019S BBB-chips with MEK1/2 inhibitor rescues barrier integrity defects. Treatment is also associated with lower levels of IL-6 and IL-8 cytokine secretion. Do blocking antibodies against IL-6 or IL-8 have any effect on barrier integrity in untreated G2019S BBB chips?

Re: We appreciate this suggestion and we accordingly provide new data showing that addition of an IL-8 neutralizing antibody to the brain compartment partially rescues dextran permeability in the BBB^{G2019S} chip (**new**

Figure 4I). This rescue effect of barrier integrity is only partial, suggesting that other factors could act synergistically to induce the observed detrimental effect. This finding is in line with our observation that vessel function is restored when inhibiting MEK/ERK signaling, a kinase cascade known to regulate the expression of diverse sets of molecules.

Minor concerns, Comment #10

It is again unusual to compute SEM or t-tests with only 2 measurements per group as in 5B.

Re: Additional replicates have been added to the experiments to strengthen our statistical analyses. A detailed explanation of these improvements is provided in our response to Reviewer 2, comment #3.

Comment #11

Figure 6D, BBB cells appear to have fully detached, Figure 3A,D vascular cells also appear to have detached or simply not formed a competent vessel.

Re: Figure 6E has been replaced with a more representative image. The permeability to 4.4 kDa dextran-TMRE observed in BBB^{G2019S} chips has been confirmed in a monolayer-based transwell model that does not rely on the formation of a competent vessel to measure barrier integrity (**new Figure 3E**).

Reviewer #3 (Remarks to the Author):

This manuscript uses iPSCs harboring a Parkinson's relevant mutation in LRRK2 to identify potential connections between astrocyte inflammation and blood-brain barrier dysfunction. The work represents a unique approach for using iPSC models to understand astrocyte-BMEC links in disease, and I think it could be a valuable addition to the literature. My main critiques are centered around data interpretation. First, I do not think the astrocyte phenotyping was done properly if the culture medium contained serum, which causes baseline astrocyte reactivity -- this would confound interpretations based on the genetic mutation. Second, some of the phenotypes observed in the vessel model are difficult to put into context. It is not clear if the vascular distortion that causes permeability increases is physiologically relevant and meaningful. This dampens enthusiasm for the significance of the work. These comments and others are elaborated below.

Comment #1

The experiments in Figure 1I-J and Figure 4 are interesting but need to be better contextualized. The method section of this manuscript says that after derivation, astrocytes were cultured in a "chemically defined astrocyte medium" from ScienCell. However, there is no chemically defined astrocyte medium sold by this company – rather, their medium is supplemented with 2% FBS, which has two problems. First, FBS will contain cytokines that impact the proteome array. Second, astrocytes become reactive in FBS, and this could influence their gene expression and protein secretion. It would be more appropriate to use a serum-free differentiation strategy to ascertain gene expression and protein secretion under a quiescent astrocyte phenotype in both genetic backgrounds.

Re: We appreciate the reviewer's comment and we updated the methods section of the manuscript to better reflect the composition of the astrocyte medium. This medium contains Fetal Bovine Serum, but the bovine cytokines are not detected by the ELISA or proteome arrays used in this study. These assays are developed specifically to detect human proteins. However, at the time of data collection, we had independently confirmed that complete media containing FBS does not interfere with the IL-6 and IL-8 ELISA (i.e. media control condition). This observation validates that the data collected is specific to human proteins. To address the reviewer's concerns that FBS in the medium could maintain the astrocytes in a reactive state, we provide a heatmap showing that

astrocytes generated for this study, as well as astrocytes used in the meta-analysis, express low to undetectable levels of inflammation-related markers (**new Supplementary Figure 5**). However, we agree with the reviewer that FBS-free differentiation strategies would be more representative of physiological conditions, and this need for better astrocyte differentiation protocols is actively discussed in the stem cell astrocyte community. To address this issue in future studies, work is underway to optimize a serum-free midbrain astrocyte differentiation protocol.

Comment #2

Although gene expression data in the BMEC-like cells are useful for verifying endothelial identity, the authors should also immunostain for VE-cadherin and PECAM-1 for explicit confirmation.

Re: To further characterize the BMEC-like cells, we now provide a combination of qPCR, western blot and immunofluorescence data showing expression of multiple markers, including VE-cadherin and CD31 (PECAM-1) (**new Figure 2B-E**).

Comment #3

Raw data should be included for Figure 2F-H similar to Figure 3A so that readers can properly judge the quantifications.

Re: The raw data have now been provided (see **new Supplementary Figure 3B-C**).

Comment #4

The data in Figure 3 should be accompanied by standard Transwell experiments. While I can appreciate the use of the microfluidic device to add three-dimensional elements and shear stress, the identification of monolayer gaps in Figure 3D hinders interpretation of barrier integrity outcomes. The authors would have no issues achieving a uniform monolayer on Transwell filters, and since the microfluidic chip is a non-contact model between the BMECs and astrocytes, a Transwell setup would complement these outcomes. An alternative would be if the authors could acquire human tissue or mouse tissue from a Parkinson's model to show similar enlargement of vessels with low cell coverage. Otherwise, the physiological relevance of these outcomes is unclear.

Re: We thank the reviewer for these suggestions which we have all incorporated to the revised version of the manuscript. In a substantial new set of experiments, we undertook both transwell and post-mortem studies. Firstly, we include a transwell experiment (**New Figure 3E**). We found that mutant astrocytes affect the permeability of a 4.4 kDa dextran in the transwell setting and this observation confirms that LRRK2 G2019S astrocytes alter the function of BMEC-like cells. Secondly, we validated neurovascular impairments in patients with PD by quantifying vessel coverage and diameter in postmortem sections of the *substantia nigra*. The new data confirm previously published observations of reduced coverage and decreased number of blood vessels¹⁵, as well as an enlargement of vessel diameter in PD patients (**New Figure 8**). This observation corroborates our data showing that LRRK2 G2019S astrocytes increase vessel width in the BBB-chip (**New Figure 3M**), and strengthens the translational capability of the BBB-chip model as well as the physiological relevance of the overall study.

Comment #5

Similar to my above comment, Figure 6 should include an analysis of BMEC coverage to mirror the phenotype observed in Figure 3.

Re: We now provide a quantification of vessel width in the **new Figure 6K**, and the data shows that ERK inhibition rescues vessel enlargement observed in the BBB^{G2019S} chips.

Comment #6

Does MEK/ERK inhibition have any effect on the BMECs alone, with or without the G2019S mutation?

Re: To address this comment, we quantified P_{aap} values for BBB^{CTL} in the presence or absence of the ERK inhibitor. The graph below shows that ERK treatment does not affect BBB^{CTL} vessel permeability. We have not included this data in the revised manuscript, but we can add it to the Supplementary Figures if requested by the reviewers.

References

1. de Rus Jacquet A, Tancredi, J.L., Lemire, A.L., DeSantis, M.C., Li, W-P., O'Shea, E. The LRRK2 G2019S mutation alters astrocyte-to-neuron communication via extracellular vesicles and induces neuron atrophy in a human iPSC-derived model of Parkinson's disease. *eLife* 2021;10:e73062.
2. de Vries HE, Blom-Roosemalen MC, van Oosten M, et al. The influence of cytokines on the integrity of the blood-brain barrier in vitro. *J Neuroimmunol* 1996;64:37-43.
3. Wong D, Dorovini-Zis K, Vincent SR. Cytokines, nitric oxide, and cGMP modulate the permeability of an in vitro model of the human blood-brain barrier. *Experimental neurology* 2004;190:446-55.
4. Szekanecz Z, Koch AE. Mechanisms of Disease: angiogenesis in inflammatory diseases. *Nat Clin Pract Rheumatol* 2007;3:635-43.
5. Kim H, Leng K, Park J, et al. Reactive astrocytes transduce inflammation in a blood-brain barrier model through a TNF-STAT3 signaling axis and secretion of alpha 1-antichymotrypsin. *Nat Commun* 2022;13:6581.
6. Nishitsuji K, Hosono T, Nakamura T, Bu G, Michikawa M. Apolipoprotein E regulates the integrity of tight junctions in an isoform-dependent manner in an in vitro blood-brain barrier model. *The Journal of biological chemistry* 2011;286:17536-42.
7. Daneman R, Agalliu D, Zhou L, Kuhnert F, Kuo CJ, Barres BA. Wnt/beta-catenin signaling is required for CNS, but not non-CNS, angiogenesis. *Proceedings of the National Academy of Sciences of the United States of America* 2009;106:641-6.
8. de Rus Jacquet A. Preparation and Co-Culture of iPSC-Derived Dopaminergic Neurons and Astrocytes. *Curr Protoc Cell Biol* 2019;85:e98.
9. Lippmann ES, Azarin SM, Palecek SP, Shusta EV. Commentary on human pluripotent stem cell-based blood-brain barrier models. *Fluids Barriers CNS* 2020;17:64.
10. Brockmann K, Schulte C, Schneiderhan-Marra N, et al. Inflammatory profile discriminates clinical subtypes in LRRK2-associated Parkinson's disease. *Eur J Neurol* 2017;24:427-e6.
11. Cook DA, Kannarkat GT, Cintron AF, et al. LRRK2 levels in immune cells are increased in Parkinson's disease. *NPJ Parkinsons Dis* 2017;3:11.
12. Ahmadi Rastegar D, Hughes LP, Perera G, et al. Effect of LRRK2 protein and activity on stimulated cytokines in human monocytes and macrophages. *NPJ Parkinsons Dis* 2022;8:34.
13. Panagiotakopoulou V, Ivanyuk D, De Cicco S, et al. Interferon-gamma signaling synergizes with LRRK2 in neurons and microglia derived from human induced pluripotent stem cells. *Nat Commun* 2020;11:5163.
14. Reinhardt P, Schmid B, Burbulla LF, et al. Genetic correction of a LRRK2 mutation in human iPSCs links parkinsonian neurodegeneration to ERK-dependent changes in gene expression. *Cell Stem Cell* 2013;12:354-67.
15. Guan J, Pavlovic D, Dalkie N, et al. Vascular degeneration in Parkinson's disease. *Brain Pathol* 2013;23:154-64.
16. Al-Bachari S, Naish JH, Parker GJM, Emsley HCA, Parkes LM. Blood-Brain Barrier Leakage Is Increased in Parkinson's Disease. *Front Physiol* 2020;11:593026.
17. Kortekaas R, Leenders KL, van Oostrom JC, et al. Blood-brain barrier dysfunction in parkinsonian midbrain in vivo. *Ann Neurol* 2005;57:176-9.
18. Gray MT, Woulfe JM. Striatal blood-brain barrier permeability in Parkinson's disease. *Journal of cerebral blood flow and metabolism : official journal of the International Society of Cerebral Blood Flow and Metabolism* 2015;35:747-50.
19. Pienaar IS, Lee CH, Elson JL, et al. Deep-brain stimulation associates with improved microvascular integrity in the subthalamic nucleus in Parkinson's disease. *Neurobiology of disease* 2015;74:392-405.
20. Pisani V, Stefani A, Pierantozzi M, et al. Increased blood-cerebrospinal fluid transfer of albumin in advanced Parkinson's disease. *J Neuroinflammation* 2012;9:188.

21. Desai Bradaric B, Patel A, Schneider JA, Carvey PM, Hendey B. Evidence for angiogenesis in Parkinson's disease, incidental Lewy body disease, and progressive supranuclear palsy. *J Neural Transm (Vienna)* 2012;119:59-71.
22. Ohlin KE, Francardo V, Lindgren HS, et al. Vascular endothelial growth factor is upregulated by L-dopa in the parkinsonian brain: implications for the development of dyskinesia. *Brain* 2011;134:2339-57.
23. Sweeney MD, Kisler K, Montagne A, Toga AW, Zlokovic BV. The role of brain vasculature in neurodegenerative disorders. *Nat Neurosci* 2018;21:1318-31.
24. Fujita K, Peng S, Ma Y, et al. Blood-brain barrier permeability in Parkinson's disease patients with and without dyskinesia. *Journal of neurology* 2021;268:2246-55.
25. Wada K, Arai H, Takanashi M, et al. Expression levels of vascular endothelial growth factor and its receptors in Parkinson's disease. *Neuroreport* 2006;17:705-9.
26. Janzer RC, Raff MC. Astrocytes induce blood-brain barrier properties in endothelial cells. *Nature* 1987;325:253-7.
27. Heithoff BP, George KK, Phares AN, Zuidhoek IA, Munoz-Ballester C, Robel S. Astrocytes are necessary for blood-brain barrier maintenance in the adult mouse brain. *Glia* 2021;69:436-72.
28. Haseloff RF, Blasig IE, Bauer HC, Bauer H. In search of the astrocytic factor(s) modulating blood-brain barrier functions in brain capillary endothelial cells in vitro. *Cell Mol Neurobiol* 2005;25:25-39.
29. Abbott NJ, Ronnback L, Hansson E. Astrocyte-endothelial interactions at the blood-brain barrier. *Nat Rev Neurosci* 2006;7:41-53.

REVIEWER COMMENTS

Reviewer #1 (Remarks to the Author):

The authors have strengthened the manuscript by addressing most of our previous comments as follows:

Major issues, comment 1-5:

No further comments.

Additional comments, comment 6-8:

No further comments.

Additional comments, comment 9:

We thank the authors for adding the characterization of iPSC-derived astrocytes from all lines to the manuscript. In order to make it easier for the reader to follow, consistent labelling of line 1, 2 and 3 (or isogenic vs non-isogenic line 1 and 2) should be used throughout the manuscript.

Additional comments, comment 10:

For Figure 2, the authors should display individual data points for the graphs.

Additional comments, comment 11-14:

No further comments.

Additional comments, comment 15:

We greatly appreciate that the authors obtained post-mortem tissue and analysed morphological changes of the brain vasculature in PD. For Figure 8, the authors should display individual data points for the graphs. Moreover, in the abstract, the authors claim that they 'confirm that these astrocyte-mediated vascular changes are also observed in the post-mortem substantia nigra of people with PD.' While this study supports that astrocytes with the LRRK2 G2019S mutation are pro-inflammatory and fail to support the formation of a functional capillary, the claims about the human findings need to be more carefully worded throughout the manuscript (e.g. line 547), in particular without characterising potential changes in astrocytes, inflammation or angiogenesis in the post-mortem tissue. Furthermore, in post-mortem studies t-tests are usually accompanied by additional statistical analysis (e.g. checking for potential co-variables or relationship to treatments etc.). Since the group size is small and the authors lack information regarding PMI, this should be okay. However, this should be addressed in the discussion.

Additional comments, comment 16:

No further comment.

Additional comments, comment 17:

Supplementary Figure 2C only shows the LRRK2 protein levels for two control and two LRRK2 lines. The authors should add the results for the third line. Moreover, the blot doesn't provide a detailed labelling (isogenic vs non-isogenic pair).

Minor issue, comment 18:

No further comment.

Reviewer #2 (Remarks to the Author):

Overall this study is greatly improved by the addition of new lines and the corollary analysis of the post mortem tissue.

Major Issues:

1. The inclusion of additional lines adds to the overall strength of this work however these new lines were not consistently included with the analysis of the isogenic pair but rather presented separately (ie Figures 4, 5F-K, 6 ect). For the main text figures the authors should present aggregate data (as done in Figure 3 B,C) from all of the LRRK2 and control lines to strengthen the conclusions of the work and limit random line to line variability.

2. The conclusion of larger BMEC cell size leading to larger vessel diameter remains insufficiently supported. While the basic characterization of the BMECs is presented in the presence of either the control or PD astrocytes (Figure 3), analysis of differential populations arising from exposure to PD specific cytokines is not shown. A non BMEC population arising in the presence of the PD signaling could be a confounding interpretation of the vessel diameter data presented Figure 3. At minimum, counter staining the vasculature chambers for contaminating neural progenitor markers such as Nestin should be included as well as a western for quantification.

Minor points:

1. The use of GO enrichment analysis vs GSEA in the figure 1 metanalysis. Since enrichment analysis of discrete categories (such as DAVID) is more likely to find false positive signals, it would strengthen the conclusions of this study if the authors validated this metanalysis using the alternative GSEA methodology. Do similar angiogenic and immune categories arise?
2. Rather than normalizing the ELISA data in Figure 5 H-K it would be more informative to present the actual pg/mL concentration levels. This would then match the data in figure 6 A-D.
3. Sup Figure 3B contains images of dye penetration but the figure legend states "Statistical analysis in (B) was performed using a two-way ANOVA with Šidák's multiple comparisons test (**** $p < 0.0001$)"
4. The nomenclature of BBBcontrol or BBBG2019S is confusing as it is not the barrier forming cells that are derived from these genetic backgrounds but the astrocytes in the other chamber.

Reviewer #3 (Remarks to the Author):

My technical concerns have been addressed. I applaud the amount of work the authors performed to strengthen this manuscript.

Reviewer #1 (Remarks to the Author):

The authors have strengthened the manuscript by addressing most of our previous comments as follows:

Major issues, comment 1-5:

No further comments.

Additional comments, comment 6-8:

No further comments.

Additional comments, comment 9:

We thank the authors for adding the characterization of iPSC-derived astrocytes from all lines to the manuscript. In order to make it easier for the reader to follow, consistent labelling of line 1, 2 and 3 (or isogenic vs non- isogenic line 1 and 2) should be used throughout the manuscript.

Re: As suggested, we updated the figures to make sure our readers can easily follow the experimental design and the use of isogenic and non-isogenic lines. In cases where several lines are combined into a single graph (e.g. Supplementary Figure 2C), a description of the lines used has been included in the figure legend.

Additional comments, comment 10:

For Figure 2, the authors should display individual data points for the graphs.

Re: We updated the figures to show individual data points for graphs with a sample size less than 10, and we now show violin plots with individual data points for larger sample sizes.

Additional comments, comment 11-14:

No further comments.

Additional comments, comment 15:

We greatly appreciate that the authors obtained post-mortem tissue and analysed morphological changes of the brain vasculature in PD. For Figure 8, the authors should display individual data points for the graphs. Moreover, in the abstract, the authors claim that they 'confirm that these astrocyte-mediated vascular changes are also observed in the post-mortem substantia nigra of people with PD.' While this study supports that astrocytes with the LRRK2 G2019S mutation are pro-inflammatory and fail to support the formation of a functional capillary, the claims about the human findings need to be more carefully worded throughout the manuscript (e.g. line 547), in particular without characterising potential changes in astrocytes, inflammation or angiogenesis in the post-mortem tissue. Furthermore, in post-mortem studies t-tests are usually accompanied by additional statistical analysis (e.g. checking for potential co-variates or relationship to treatments etc.). Since the group size is small and the authors lack information regarding PMI, this should be okay. However, this should be addressed in the discussion.

Re: The graphs in Figure 8 have been updated to better reflect data distribution, as described in our response to comment 10. In addition, we updated the text throughout the manuscript to provide a more cautious analysis of the human data and ensure that we do not overstate our findings. We also agree with the reviewer that our

discussion would be improved by the mention of the additional statistical analyses that could be performed with a larger sample size. The manuscript has been modified using the “track-changes” option, and we provide an example of the new text below (changes are highlighted in blue):

By modeling the complex brain vasculature *in vitro*, we identified astrocyte-related pathological changes to vessel function and morphology. Postmortem analysis of PD vs. control *substantia nigra* tissue corroborated our observations of vessel enlargement [*astrocyte-specific text removed*] (Figure 8). The investigation of postmortem brain samples revealed additional pathological modifications to the complexity of the brain vasculature in the *substantia nigra* of PD patients, as demonstrated by changes to the number and total coverage area of blood vessels. These findings are consistent with an earlier study that suggested decreased blood vessel density and increased vessel diameter in the postmortem brain of PD patients¹, and with an alpha-synuclein overexpression *in vivo* model that documented BBB leakage associated with lower striatal vessel density compared to wild-type animals². However, the specific role of astrocytes in human vascular pathology should be addressed in future studies, for example by correlating postmortem quantification of vascular changes with astrocyte reactivity and angiogenic potential. In addition, further exploring these changes in a larger cohort of patients would enable in-depth statistical analyses that consider the impact of covariates such as comorbidities, treatment regimen, or biological sex.

Additional comments, comment 16:

No further comment.

Additional comments, comment 17:

Supplementary Figure 2C only shows the LRRK2 protein levels for two control and two LRRK2 lines. The authors should add the results for the third line. Moreover, the blot doesn't provide a detailed labelling (isogenic vs non-isogenic pair).

Re: We apologize if the figure legend was unclear, as the original graph in Supplementary Figure 2C (now Supplementary Figure 3C) showed the combined data for all three iPSC lines. We chose this format because we did not observe significant differences between astrocyte lines for this quality control experiment. However, for clarity, we now show the data for each pair individually.

Minor issue, comment 18:

No further comment.

Reviewer #2 (Remarks to the Author):

Overall this study is greatly improved by the addition of new lines and the corollary analysis of the post mortem tissue.

Major Issues, comment #1

The inclusion of additional lines adds to the overall strength of this work however these new lines were not consistently included with the analysis of the isogenic pair but rather presented separately (ie Figures 4, 5F-K, 6 ect). For the main text figures the authors should present aggregate data (as done in Figure 3 B,C) from all of the LRRK2 and control lines to strengthen the conclusions of the work and limit random line to line variability.

Re: We appreciate that the reviewer noted how the new lines improved the manuscript and strengthened our conclusions. We updated the figures to show aggregate data instead of presenting the results separately.

Major Issues, comment #2

The conclusion of larger BMEC cell size leading to larger vessel diameter remains insufficiently supported. While the basic characterization of the BMECs is presented in the presence of either the control or PD astrocytes (Figure 3), analysis of differential populations arising from exposure to PD specific cytokines is not shown. A non BMEC population arising in the presence of the PD signaling could be a confounding interpretation of the vessel diameter data presented Figure 3. At minimum, counter staining the vasculature chambers for contaminating neural progenitor markers such as Nestin should be included as well as a western for quantification.

Re: We appreciate the reviewer’s concern, and have consequently completed additional experiments to determine if contaminating cell types are appearing in the vessel (see new Rebuttal Figure 1). When selecting markers of contaminating cells, we elected to exclude Nestin as previous studies have reported Nestin expression in endothelial cells³. We instead followed two approaches, (1) assessing the proportion of cells displaying the expected cellular localization of ZO-1 at the tight junctions (i.e. outer edges of the cells), and (2) measuring the levels of the proliferation marker Ki67. As shown in Rebuttal Figure 1A, we quantified the proportion of cells in the vascular compartment that display ZO-1 staining specifically at the tight junctions, to supplement Manuscript Figure 3H (ZO-1 western blot quantification) and Figure 3J (ZO-1 immunostaining). The data show that nearly 100 % of cells growing in the vascular compartment express ZO-1 at cell-cell junctions, thus supporting the absence of contaminating progenitors. Secondly, we evaluated the presence of proliferating cells by quantifying Ki67 levels by western blot (Rebuttal Figure 1B). The data show no differences in Ki67 levels between BBB^{G2019S} and BBB^{CTL} vessels. Overall, the data presented in Rebuttal Figure 1 and our routine observations of BBB^{G2019S} and BBB^{CTL} vessels do not indicate the presence of non-BMEC populations that may affect vessel diameter. We decided not to include the data collected for this rebuttal in the main manuscript, but we would be happy to add this supplementary information to the paper at the reviewer’s request.

Rebuttal Figure 1. BBB^{G2019S} vessels have the same cellular composition as BBB^{CTL} vessels. A. Graph showing the proportion of vascular cells expressing the tight junction marker ZO-1 at cellular junctions in BBB^{G2019S} vs. BBB^{CTL} vessels. We refer the reviewer to Figure 3J of the manuscript for representative images of ZO-1 staining, and Figure 3H for western blot quantifications of ZO-1 levels. **B.** Graph showing western blot quantification of the proliferation marker Ki67 in BBB^{G2019S} vs BBB^{CTL} vessels. The data points represent independent biological replicates.

Minor points, comment #1

The use of GO enrichment analysis vs GSEA in the figure 1 metanalysis. Since enrichment analysis of discrete categories (such as DAVID) is more likely to find false positive signals, it would strengthen the conclusions of this study if the authors validated this metanalysis using the alternative GSEA methodology. Do similar angiogenic and immune categories arise?

Re: Following the reviewer's suggestion, we implemented the GSEA methodology and observed an enrichment in immune- and angiogenesis-related categories. In addition, we also identified an enrichment in categories related to the MEK/ERK pathway. We prepared a new Supplementary Figure 2 to show a summary table and a heatmap of genes most frequently enriched in the immune- and angiogenesis-related categories.

Minor points, comment #2

Rather than normalizing the ELISA data in Figure 5 H-K it would be more informative to present the actual pg/mL concentration levels. This would then match the data in figure 6 A-D.

Re: We updated Figure 5 to show the aggregated data for both iPSC pairs used in these experiments, and the graphs now represent pg/ml cytokine levels instead of a normalization.

Minor points, comment #3

*Sup Figure 3B contains images of dye penetration but the figure legend states "Statistical analysis in (B) was performed using a two-way ANOVA with Šidák's multiple comparisons test (**** $p < 0.0001$)".*

Re: This typo has been corrected.

Minor points, comment #4

The nomenclature of BBBcontrol or BBBG2019S is confusing as it is not the barrier forming cells that are derived from these genetic backgrounds but the astrocytes in the other chamber.

Re: We understand the reviewer's concern. In addition to defining the BBB^{CTL} and BBB^{G2019S} nomenclature in Figures 2A and 3A, we added the following explanation to each relevant figure legend: "The BBB^{CTL} and BBB^{G2019S} nomenclature refers to the presence of either control or LRRK2 G2019S astrocytes in the brain compartment of the BBB-chips".

Reviewer #3 (Remarks to the Author):

My technical concerns have been addressed. I applaud the amount of work the authors performed to strengthen this manuscript.

Re: We sincerely appreciate this kind feedback, and we are happy to have improved the manuscript with the help of our reviewers.

References

- 1 Guan, J., Pavlovic, D., Dalkie, N., Waldvogel, H. J., O'Carroll, S. J., Green, C. R. & Nicholson, L. F. Vascular degeneration in Parkinson's disease. *Brain Pathol* **23**, 154-164 (2013).

- 2 Elabi, O., Gaceb, A., Carlsson, R., Padel, T., Soylu-Kucharz, R., Cortijo, I., Li, W., Li, J. Y. & Paul, G. Human alpha-synuclein overexpression in a mouse model of Parkinson's disease leads to vascular pathology, blood brain barrier leakage and pericyte activation. *Sci Rep* **11**, 1120 (2021).
- 3 Dusart, P., Fagerberg, L., Perisic, L., Civelek, M., Struck, E., Hedin, U., Uhlen, M., Tregouet, D. A., Renne, T., Odeberg, J. & Butler, L. M. A systems-approach reveals human nestin is an endothelial-enriched, angiogenesis-independent intermediate filament protein. *Sci Rep* **8**, 14668 (2018).

REVIEWERS' COMMENTS

Reviewer #2 (Remarks to the Author):

The paper has been significantly improved and have only one remain concern about the data in Figure 3B, C, and M. The authors are reporting increased apparent permeability in the G2019S BBB Chips (Fig 3B and 3C), but also increased vessel width (Fig 3M). However, it appears from the methods that the authors used the exact same surface area in their apparent permeability calculations for both conditions. This would lead to artificially inflating the apparent permeability of the larger vessels since the increase in surface area is not accounted for in the Papp calculation. The data in Fig 3E (on transwells where surface area is constant) support their claims of increased permeability in G2019S. However, it seems like the increased permeability reported in BBB chips is slightly misleading given there is no correction for the differences in vessel surface area which are reported to be significantly different. Perhaps the authors can clarify this for the editor and readers?

Reviewer #2 (Remarks to the Author):

The paper has been significantly improved and have only one remain concern about the data in Figure 3B, C, and M.

Comment #1

The authors are reporting increased apparent permeability in the G2019S BBB Chips (Fig 3B and 3C), but also increased vessel width (Fig 3M). However, it appears from the methods that the authors used the exact same surface area in their apparent permeability calculations for both conditions. This would lead to artificially inflating the apparent permeability of the larger vessels since the increase in surface area is not accounted for in the P_{app} calculation. The data in Fig 3E (on transwells where surface area is constant) support their claims of increased permeability in G2019S. However, it seems like the increased permeability reported in BBB chips is slightly misleading given there is no correction for the differences in vessel surface area which are reported to be significantly different. Perhaps the authors can clarify this for the editor and readers?

Re: The reviewer raises an important point. A constant surface area was used in all calculations related to P_{app}, even when the vessels themselves were of various width. We elected to follow this procedure because the surface area of the interface between the media and the vessel tends to remain constant regardless of the width of the vessel. As shown in the BBB-chip specifications available at https://www.mimetas.com/files/products/OrganoPlate%203-lane%2040/2023_Productflyer_3_lane_40_Specification%20sheet.pdf, the 3D vessel is bounded by either glass or proprietary polymers on three sides of the microfluidic channel (top, bottom, and side 1). The remaining side (side 2) is the vessel/ECM interface, and it is the only place where exchange between vascular dextran/rhodamine media and the brain compartment is likely to occur. When the vessel morphology changes, height is constrained by the glass/polymer, and the vessel grows further down into the semi-rigid ECM interphase without modifying the effective area defined by V_{gel} and A_{barrier} in the P_{app} equation. As a results V_{gel} and A_{barrier} remain constant across experimental conditions, even when vessel width is different. The Methods section has been updated as follows (changes highlighted in blue):

Methods - BBB-chip barrier integrity assay

Fluorescence intensity representing dye migration from the vascular to the brain compartments was quantified using ImageJ (version 1.53T) ¹ and P_{app} values were calculated using the following formula:

$$P_{app_value} \text{ (cm/s)} = (I_{end} - I_{initial}) / (T_{end} - T_{initial}) * V_{gel} / (A_{barrier}),$$

where cm = centimeter, s = seconds; I_{initial} = initial intensity; I_{end} = endpoint intensity; T_{initial} = time intial in seconds; T_{end} = time end in seconds; V_{gel} = 1.04 x 10⁴, and A_{barrier} = 5.7 x 10³. A_{Barrier} and V_{gel} were considered constants since the area available for molecule exchange was constrained by the dimensions of the plate. In these conditions, the assumption is that the fluid exchange area between the vessel and the brain compartments remains constant regardless of vessel width.

References

- 1 Rasband, W. S. *ImageJ*, <<https://imagej.nih.gov/ij>> (1997-2018).